# TEACH MULTIMODAL LLMS TO COMPREHEND ELECTROCARDIOGRAPHIC IMAGES

## ABSTRACT

The electrocardiogram (ECG) is an essential non-invasive diagnostic tool for assessing cardiac conditions. Existing automatic interpretation methods suffer from limited generalizability, focusing on a narrow range of cardiac conditions, and typically depend on raw physiological signals, which may not be readily available in resource-limited settings where only printed or digital ECG images are accessible. Recent advancements in multimodal large language models (MLLMs) present promising opportunities for addressing these challenges. However, the application of MLLMs to ECG image interpretation remains challenging due to the lack of instruction tuning datasets and well-established ECG image benchmarks for quantitative evaluation. To address these challenges, we introduce `ECGInstruct`, a comprehensive ECG image instruction tuning dataset of over 1 million samples, covering a wide range of ECG-related tasks from diverse data sources. Using `ECGInstruct`, we develop `PULSE`, a fine-tuned MLLM tailored for ECG image interpretation. In addition, we curate `ECGBench`, a new evaluation benchmark covering four key ECG image interpretation tasks. Our experiments show that PULSE sets a new state-of-the-art, outperforming general MLLMs with an average accuracy improvement of 15% to 30%. This work highlights the potential of `PULSE` to enhance ECG interpretation in clinical practice.[1]

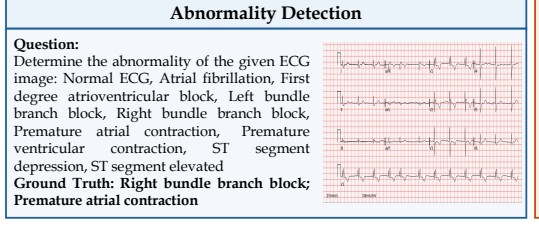

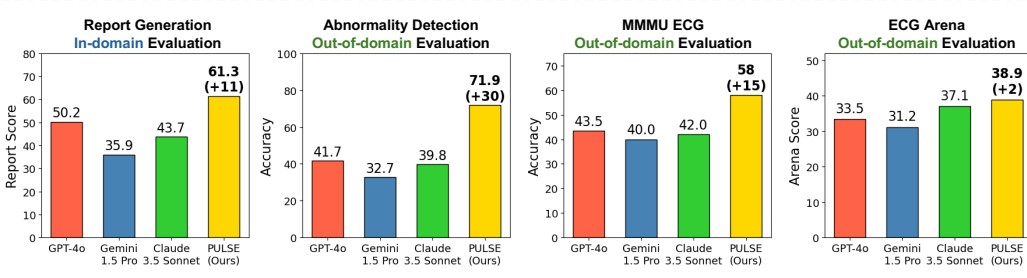

Figure 1: The proposed `PULSE` demonstrates superior performance across multiple in-domain and out-of-domain datasets on our constructed `ECGBench` compared with advanced proprietary MLLMs (e.g., GPT-4o). Notably, these proprietary MLLMs often fail to accurately interpret ECG images, generating well-structured and contextually relevant responses but ultimately incorrect (with errors highlighted in red) compared to the ground truth diagnosis.

---

[1]All code, data and models are available at `anonymous.4open.science/r/PULSE-4ECD`

# 1 INTRODUCTION

The electrocardiogram (ECG) is an essential tool in diagnosing cardiovascular diseases due to its non-invasive, cost-effective, and widely accessible nature for assessing cardiac function. While some approaches have been proposed for automatic ECG diagnosis (Hannun et al., 2019; Ribeiro et al., 2020; Hughes et al., 2021), their application in real-world clinical settings encounters several challenges. First, these models are primarily designed for classification tasks with limited cardiac conditions (Ribeiro et al., 2020), often lacking generalizability to rare or unseen abnormalities. Second, they typically treat ECG data as *time-series physiological signals*, which may not always be available, particularly in resource-constrained or remote settings (Siontis et al., 2021). In such settings, ECG data are often stored exclusively as *printed or digital images* (Sangha et al., 2022; 2023), limiting the utility of signal-based models. Third, variations in data formats and architectures across different device vendors (Cuevas-González et al., 2022) further complicate the interoperability and applicability of traditional ECG models in diverse healthcare environments (Chung et al., 2022).

Recent advancements in multimodal large language models (MLLMs) have shown impressive success across vision-language tasks, offering new possibilities for addressing the aforementioned limitations of traditional ECG models. They enable model inference directly from ECG images, which are the primary formats used by clinicians (Cuevas-González et al., 2022), and accommodating rural or remote clinic settings. However, applying MLLMs to ECG interpretation is not straightforward. As illustrated in Fig. 1, current MLLMs, such as GPT-4o (OpenAI, 2024), often provide responses that appear correct and contextually relevant but are ultimately inaccurate in interpreting ECG images. This highlights the need for specialized MLLMs for ECG image interpretation.

Developing MLLMs for ECG images faces several challenges. First, no large-scale ECG image datasets are currently available as most ECG datasets contain only raw signal data, which needs to be synthesized into digital images. Second, there is a lack of instruction tuning datasets for ECG images. Large high-quality instruction tuning datasets, which are crucial for MLLM development, need to be curated from scratch for ECG-related tasks. Finally, evaluation is just as critical as model development, yet no established benchmark exists for assessing MLLM performance in ECG image interpretation. A well-defined benchmark is essential for both quantifying model performance and identifying areas for future improvement.

In this paper, we tackle these challenges by introducing `ECGInstruct`, the first large-scale ECG image instruction tuning dataset containing over one million ECG image-text samples. `ECGInstruct` is characterized by: 1) realistic image synthesis that replicates artifacts commonly seen in paper-based ECGs, 2) a diverse range of ECG-related tasks with clinical experts' insights for refinement, and 3) data sourced from distinct geographic regions. We use `ECGInstruct` to fine-tune LLaVA (Liu et al., 2024b), resulting in our model, `PULSE`. To address the evaluation challenge, we present `ECGBench`, a comprehensive evaluation benchmark covering four major ECG image analysis tasks. `ECGBench` includes repurposed diagnosis and clinical report generation tasks from existing datasets, as well as newly created, complex ECG analysis using real-world images.

Evaluated on `ECGBench`, `PULSE` sets a new state-of-the-art, significantly outperforming proprietary MLLMs across all benchmarks with an average accuracy gain of 15% to 30% compared to GPT-4o on out-of-domain datasets (Fig. 1). Ablation experiments demonstrate the importance of incorporating diverse data sources and ECG instruction tasks into the training data. A case study and discussion further illustrate the model's effectiveness in ECG image interpretation.

To summarize, our main contributions are as follows,

• **Problem.** We investigate the capabilities of MLLMs in ECG image interpretation and evaluate their performance across various downstream tasks. To the best of our knowledge, this is the first study focused on assessing MLLMs in image-based ECG interpretation.

• **Dataset.** We construct `ECGInstruct`, a large-scale ECG image instruction tuning dataset consisting of various ECG-related tasks, serving as a valuable resource for fine-tuning MLLMs.

• **Model.** We develop `PULSE`, a new MLLM tailored for ECG image interpretation. The model achieves state-of-the-art performance, outperforming both proprietary and open-source MLLMs.

• **Evaluation.** We establish `ECGBench`, a comprehensive benchmark for evaluating ECG image interpretation, which includes diverse evaluation tasks, both real-world and synthesized images.

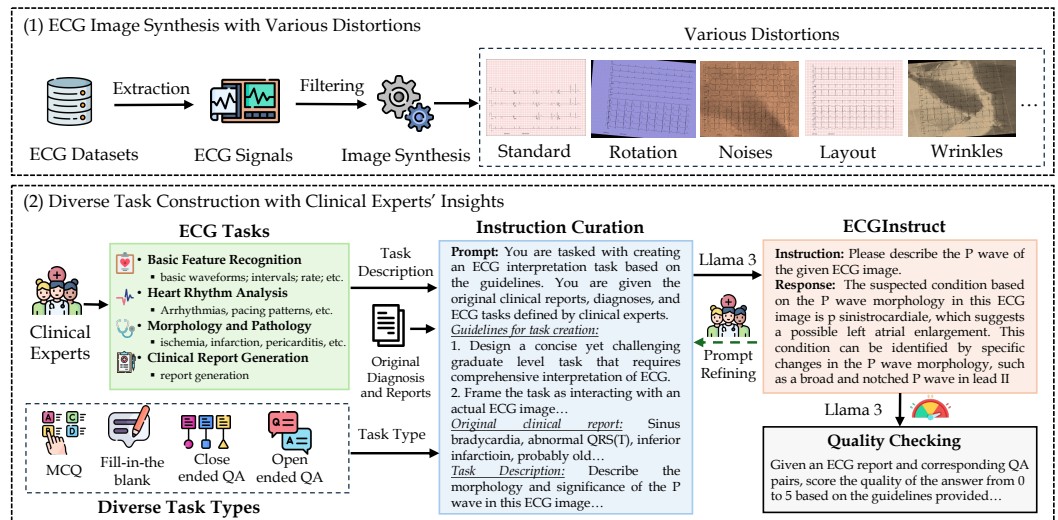

Figure 2: `ECGInstruct`: a list of diverse and large-scale instruction tuning datasets for ECG image interpretation. (1) ECG images are synthesized from raw signal recordings with various distortions that mimic real-world printed ECG images. (2) `ECGInstruct` is curated based on clinician-defined ECG-related tasks, original diagnosis and clinical reports, and diverse task types. Additional quality checking is applied to filter lower-scored instructions.

| Source Dataset | Task | Type | # Samples |
|---|---|---|---|
| PTB-XL (Wagner et al., 2020) | Feature | Close/Open/Fill/MCQ | 30K |
| | Rhythm | Close/Open/Fill/MCQ | 36K |
| | Morphology | Close/Open/Fill/MCQ | 67K |
| | Report | Open | 16K |
| ECG-QA (Oh et al., 2024) | Feature | Close | 40K |
| | Rhythm | Close | 9K |
| | Morphology | Close | 90K |
| MIMIC-IV-ECG (Gow et al., 2023) | Feature | Close/Open/Fill/MCQ | 29K |
| | Rhythm | Close/Open/Fill/MCQ | 115K |
| | Morphology | Close/Open/Fill/MCQ | 169K |
| | Report | Open | 487K |
| CODE-15% (Ribeiro et al., 2021) | Feature | Close | 22K |
| | Rhythm | Close | 14K |
| | Morphology | Close | 31K |
| Total (`ECGInstruct`) | | | 1.2M |

Table 1: Summary of `ECGInstruct`. Feature: basic feature recognition, Rhythm: heart rhythm analysis, Morphology: morphology and pathology identification, Report: clinical report generation. Close: close-ended QA, Open: open-ended QA, Fill: fill-in-the-blank, MCQ: multi-choice QA. The full table of data statistics is provided in Appendix Table A1.

## 2 ECGINSTRUCT: TEACH MLLMS TO COMPREHEND ECG IMAGES

We aim to curate a list of multifaceted instruction tuning datasets for ECG analysis that are featured by 1) realistic image synthesis resembling the artifacts in paper ECGs, 2) diverse types of ECG-related tasks with clinical experts' insights, and 3) different data sources from distinct geographical regions. We show the construction of `ECGInstruct` in Fig. 2 and data summary in Table 1.

**ECG Image Synthesis with Real Distortions.** To enhance the robustness and real-world applicability of our model, we synthesize ECG images mimicking common artifacts found in paper ECGs. We adopt an ECG image synthesis tool (Shivashankara et al., 2024) that provides various imper-

fections such as grid line interference, creases, wrinkles, paper rotations, etc. By including these synthesized artifacts, we aim to train models that can effectively interpret ECGs in less-than-ideal conditions, as often encountered in clinical settings. More details are provided in Appendix C.

**ECG-related Tasks with Clinical Experts' Insights.** To construct a comprehensive set of ECG-related tasks, we consulted domain experts to curate diverse and clinically relevant tasks covering four different categories. Each category is designed to address specific aspects of ECG interpretation and analysis, including (1) basic feature recognition (see examples in Appendix Fig. A1), (2) heart rhythm analysis (see examples in Appendix Fig. A2), (3) morphology and pathology identification (see examples in Appendix Fig. A3) and (4) clinical report generation (see examples in Appendix Fig. A4). Basic feature recognition (e.g., interval or segment, etc.) forms the foundation of ECG interpretation, enabling the model to grasp essential cardiac parameters. Heart rhythm analysis (e.g., arrhythmias, conduction abnormalities, etc.) and morphology and pathology identification (e.g., wave shape, pathological conditions, etc.) are more advanced and critical aspects of ECG analysis, ensuring that the model can detect and classify complex conditions accurately. Lastly, clinical report generation mirrors the process of healthcare professionals synthesizing a comprehensive interpretation of an ECG. By incorporating clinical experts' insights, we encourage the model to learn the practical skills required in a clinical context.

**Diverse Types of Tasks.** Based on the original diagnoses and clinical reports from the existing ECG datasets, we curate diverse types of tasks including multi-choice questions, fill-in-the-blank, close-ended QA, and open-ended QA. This variety of task types not only enhances the model's versatility but also mimics the diverse cognitive processes involved in real-world ECG interpretation. By incorporating these varied task types, we aim to develop a more robust and adaptable model capable of handling a wide spectrum of ECG-related queries and analyses.

**Diverse Data Sources from Distinct Regions.** To ensure broad applicability and generalizability, we collect ECG data from four different sources across geographically distinct regions: 1) PTB-XL (Wagner et al., 2020): a Germany-based, publicly available repository; (2) MIMIC-IV-ECG (Gow et al., 2023): a large set of ECGs for patients who appear in the MIMIC-IV Clinical Database from Beth Israel Deaconess Medical Center in Boston (Johnson et al., 2023); 3) CODE-15% (Ribeiro et al., 2021): an ECG dataset from a central ECG repository from Minas Gerais, Brazil under the clinical outcomes in digital electrocardiology (CODE) study (Ribeiro et al., 2019); 4) ECG-QA (Oh et al., 2024), a question answering dataset for ECGs that is constructed based on PTB-XL (Wagner et al., 2020). This diverse geographical representation enhances the model's ability to generalize across different populations and healthcare systems, accounting for potential variations in ECG patterns and interpretations across regions.

**Data Synthesizing at Scale.** Since large-scale annotation of ECG features is extremely expensive and time-consuming, we develop an automatic data synthesizing pipeline to address this data scarcity issue. We utilized diagnostic reports from PTB-XL and MIMIC-IV-ECG as initial seed data and leveraged an advanced large language model (i.e., Llama-3-70B-Instruct) for data synthesis. Building upon the expert-in-the-loop process and diverse data resources described in the previous sections, we synthesized a substantial volume of ECG-related instructions and corresponding responses. These were based on expert-provided examples and real-world scenarios, with the specific prompts used in this process detailed in the Appendix E. For datasets lacking comprehensive reports, such as CODE-15%, we manually constructed diverse templates to transform the existing data into an instruction-response format.

**Quality Control.** To guarantee the quality of generated instructions and corresponding responses, we apply an independent LLM as a judge to evaluate and score the content. This process involves several steps: 1) initial generation: instructions and responses are first generated using our primary model; 2) evaluation criteria: we establish a set of evaluation criteria including the instruction relevance, clarity, answerability of the responses, etc; 3) LLM judge and scoring: an independent LLM (Llama 3 (Meta, 2024)) is used as a judge to assess each instruction-response pair against established criteria and assign scores (see prompt in Appendix Fig. A8); 4) feedback loop: low-scoring items are flagged for human expert review and potential revision or removal; 5) iterative refinement: based on the scoring patterns and human expert input, we continually refine our instruction generation process. By combining automated LLM evaluation with human expert oversight, we create a robust system for maintaining and improving the quality of our instruction-response pairs.

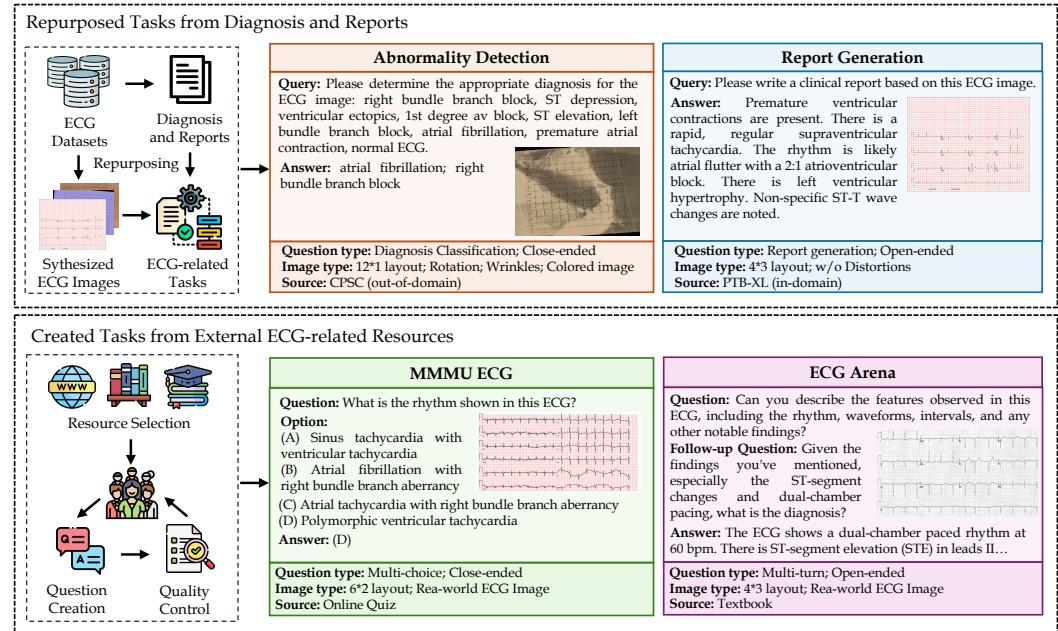

Figure 3: The data curation process for `ECGBench`. There are four key tasks involved: 1) two repurposed tasks (abnormality detection and report generation) derived from existing ECG datasets, where ECG images are synthesized from raw signals, and queries/answers are extracted based on diagnostic and clinical reports; 2) Two newly developed tasks using external resources, where ECG images and associated questions and answers are collected and generated from real-world sources.

**Training.** Our model architecture closely follows that of LLaVA (Liu et al., 2024b;c), adapting it for ECG image analysis. We use a vision encoder to process ECG images and a large language model as the text decoder, connected via a projection layer. We organize the data into three components: the image, the instructions, and the outputs. The instruction is query or task related to the ECG image and the output is the expected response or prediction base on the image and instruction. We place the image at the beginning of each conversation, serving as the visual grounding for the entire dialogue. During training, we freeze the parameters of the vision encoder while updating the parameters of the projection layer and the language model using an autoregressive training objective, where we mask all the tokens belonging to the image and the instruction.

## 3 ECGBENCH

In this section, we present ECG-Bench (Fig. 3), a comprehensive benchmark for evaluating MLLMs on ECG image interpretation. Our benchmark contains both repurposed tasks from six existing datasets and newly created tasks from external resources. Table 2 shows the details of each evaluation dataset. We introduce the detailed evaluation task curation process below.

### 3.1 EVALUATION TASK CURATION

**Abnormality Detection.** This task focuses on detecting cardiac abnormalities using ECG images. We curate this task by repurposing six existing ECG datasets: three in-domain datasets: PTB-XL (Super) (Wagner et al., 2020), CODE-15% (Ribeiro et al., 2021), ECG-QA (Oh et al., 2024), and three out-of-domain datasets: CPSC 2018 (Liu et al., 2018), CSN (Zheng et al., 2020a;b) and G12EC (Liu et al., 2018). For all datasets, we first synthesize images using raw signals and then curate queries based on the original diagnosis and reports. For datasets with fewer than 10 diagnostic labels, we curate close-ended questions. For those with more labels, we construct multi-choice questions with 8 options, including the original diagnosis and randomly sampled negative labels.

**Report Generation.** This task involves generating detailed reports for given ECG images. We benchmark using 500 randomly selected reports from the test set of PTB-XL, which contains high-

| Evaluation Dataset | Task | Type | # Samples | In-Domain? |
|---|---|---|---|---|
| PTB-XL Super | Abnormality Detection | Close-ended | 2,082 | YES |
| PTB-XL Report | Report Generation | Open-ended | 500 | YES |
| CODE-15% | Abnormality Detection | Close-ended | 1,400 | YES |
| ECG-QA | Abnormality Detection | Close-ended | 1,317 | YES |
| CPSC 2018 | Abnormality Detection | Close-ended | 2,061 | NO |
| CSN | Abnormality Detection | MCQ (8-option) | 1,611 | NO |
| G12EC | Abnormality Detection | MCQ (8-option) | 2,026 | NO |
| MMMU ECG | Multimodal Understanding | MCQ (4-option) | 200 | NO |
| ECG Arena | Multi-turn Conversation | Open-ended | 50 | NO |

Table 2: Overview of evaluation datasets in `ECGBench`. This collection contains both in-domain and out-of-domain problems across four key tasks with diverse answer types.

quality ECG reports written and validated by cardiologists. Similarly, the ECG images are synthesized from the raw signals. For the ground truth reports written in non-English (PTB-XL is a Germany-based dataset), we translate the reports into English before the evaluation.

**MMMU ECG.** Inspired by MMMU (Yue et al., 2024), a widely adopted evaluation benchmark for MLLMs, we manually curated an ECG version with 200 multi-choice questions with the help of medical school students. The curation process involved three key steps: (1) **Resource Selection:** We gathered ECG materials from diverse and reliable sources such as ECG textbooks, clinical case reports from medical journals, and widely used online ECG learning materials. This ensures the comprehensiveness and quality of collected ECG examples and interpretations. (2) **Question Creation and Collection:** Five medical school students with basic knowledge of ECG were recruited for this task. They extracted existing questions from the collected resources. For ECG images accompanied only by clinical interpretations, the annotators created questions based on these interpretations. Additionally, they formulated new questions drawing from their expertise, ensuring a balance between various ECG interpretation aspects (e.g., rhythm analysis, morphology assessment, clinical interpretation). (3) **Quality Control:** To maintain high standards, we implemented a quality control process. In particular, Each question underwent review by at least two other annotators, checking for accuracy and clarity. An independent reviewer cross-checked the final images, questions, and corresponding answers against the original sources to ensure fidelity to the source material. Any discrepancies or ambiguities were resolved during this process.

**ECG Arena.** To assess the model's instruction-following ability in ECG comprehension, we developed ECG Arena, inspired by MT-Bench (Zheng et al., 2024) and Arena-hard (Chiang et al., 2024) used in general LLM chat evaluations. We manually curated 50 multi-turn ECG-related questions, focusing on open-ended interactions. The data curation process for ECG Arena, like MMMU ECG, involves three main steps: resource selection, question creation, and quality control. The key distinction is that MMMU ECG focuses on multiple-choice questions, whereas ECG Arena involves more complex, flexible multi-turn, open-ended questions. Each follow-up question is contingent on the initial question and its response, making the process more challenging and reflective of real-world applications. Since multi-turn conversations are rare in existing sources, this posed significant challenges during data curation. To address this, annotators created such conversations by referencing original clinical interpretations and ECG images. The questions are designed to feel natural and simulate a real clinical setting (e.g., the first question may ask about basic findings from the image, followed by a question about potential clinical causes or diagnoses based on those findings).

## 3.2 EVALUATION METRICS.

**Abnormality Detection:** We use macro AUC, macro F1, and hamming loss (HL) for multi-label datasets, and accuracy for others. **Report Generation:** We employ GPT-4o as a judge, evaluating reports based on rhythms, waveform, and diagnosis, with a maximum score of 100 points (see evaluation prompt in Appendix Fig. A9). **MMMU ECG:** We use accuracy as the primary metric, with systematic, rule-based evaluation pipelines to ensure consistent scoring. **ECG Arena:** GPT-4o assesses model performance by comparing generated responses with ground truth answers, considering accuracy, completeness, and instruction adherence, with a maximum score of 100 points (see

evaluation prompt in Appendix Fig. A10). More evaluation details are provided in the Appendix F.1.

## 4 EXPERIMENTS

### 4.1 METHODS FOR COMPARISON

In order to evaluate the performance of our proposed model, we compare it against a set of established methods including domain-specific methods and state-of-the-art MLLMs.

- **Domain-specific Methods:** We consider four domain-specific methods for ECG including three signal-based methods: METS (Li et al., 2024c), MERL (Liu et al., 2024a), ST-MEM (Na et al., 2023), and one image-based method: ECG-GPT (Khunte et al., 2024).
- **Proprietary MLLMs:** We consider three proprietary MLLMs: GPT-4o, GPT-4o mini (OpenAI, 2024), Gemini 1.5 Pro (Reid et al., 2024) and Claude 3.5 Sonnet (Anthropic, 2024).
- **Open-source MLLMs:** We select a range of open-source models to ensure comprehensive coverage across different model sizes and visual components, including the general models LLaVA-1.5 (Liu et al., 2024d;b), LLaVA-1.6 (Liu et al., 2024c), Phi-3-Vision Abdin et al. (2024), Idefics2-8B (Laurençon et al., 2024), DeepSeek-Vl-7B (Lu et al., 2024a), Mantis-8B-siglip-Llama3 (Jiang et al., 2024), MiniCPM-V-2.6 (Yao et al., 2024), InternVL2 (Chen et al., 2023; 2024) and state-of-the-art multimodal models LLaVA-OneVision (Li et al., 2024a), Qwen2-VL (Wang et al., 2024), as well as the domain-specific models LLaVA-Med (Li et al., 2024b).

### 4.2 IMPLEMENTATION DETAILS

We follow the architecture of LLaVA-v1.6-Vicuna-7B, which includes three core components: a vision encoder, a large language model, and a projector to align image and text modalities. We format all datasets into a chatbot-style multi-turn dialogue format and use the special token "<image>" to represent image features within the text data. Detailed implementation details are provided in Appendix F.2.

### 4.3 MAIN RESULTS

We show in-domain the out-of-domain results in Table 3 and Table 4 respectively. Overall, we observe that PULSE achieves state-of-the-art performance on different datasets and tasks.

**Results on In-domain datasets.** As shown in Table 3, PULSE demonstrates significant improvements over both proprietary and open-source MLLMs across all in-domain datasets. Specifically, PULSE surpasses the best proprietary model (GPT-4o) with a 27% improvement in AUC, an 11-point gain in report score, and a 39% increase in accuracy on the PTB-XL Super, PTB-XL Report, and ECG-QA tasks, respectively. Moreover, PULSE achieves notable gains over the best open-source model, with a 28% improvement in AUC, a 12-point gain in report score, and a 44% increase in accuracy on the same tasks.

These results highlight the complexity of ECG image interpretation, a task where even the best proprietary models perform near randomly. By fine-tuning on ECGInstruct, PULSE achieves substantial performance improvements, demonstrating the importance of high-quality and task-related instruction tuning. Moreover, while certain domain-specific methods (e.g., MERL) achieve comparable performance on specific datasets, their specialized designs limit their generalization to other diverse tasks, restricting their broader applicability in real-world, complex healthcare scenarios.

**Results on Out-of-domain datasets.** Table 4 presents the comparison results on out-of-domain datasets, where PULSE consistently delivers outstanding performance. Notably, it achieves a significant 15% improvement in accuracy on the MMMU ECG benchmark compared to GPT-4o. This substantial improvement indicates the PULSE's robustness and ability to generalize to unseen data.

The ECG Arena benchmark presents a significantly more challenging task for all models. This benchmark is characterized by its multi-turn, open-ended question-answering format, which closely simulates real clinical scenarios. Despite these challenges, PULSE still surpasses the best proprietary model by 2 points and outperforms the leading open-source model by an impressive 11 points

| Datasets | PTB-XL Super | | | PTB-XL Report | CODE-15% | | | ECG-QA |
|---|---|---|---|---|---|---|---|---|
| Metric | AUC | F1 | HL | Report Score | AUC | F1 | HL | Accuracy |
| Random | 50.3 | 33.2 | 50.1 | 0 | 48.8 | 15.0 | 32.1 | 16.2 |
| *Domain-specific Methods* | | | | | | | | |
| METS | - | 65.7$^\dagger$ | - | N/A | - | - | - | N/A |
| MERL | 74.2$^\dagger$ | - | - | N/A | - | - | - | N/A |
| ST-MEM | 71.4$^\dagger$ | - | - | N/A | - | - | - | N/A |
| ECG-GPT | 69.5* | 53.9* | 20.1* | 47.8* | 68.9* | 40.1* | 17.4* | N/A |
| *Proprietary MLLMs* | | | | | | | | |
| GPT-4o | 55.6 | 28.3 | 26.2 | 50.2 | 59.9 | 24.9 | 15.7 | 35.2 |
| GPT-4o mini | 52.0 | 20.4 | 31.7 | 37.1 | 57.5 | 22.0 | 15.1 | 14.9 |
| Gemini 1.5 Pro | 50.7 | 15.3 | 27.9 | 35.9 | 56.7 | 20.0 | 15.9 | 33.2 |
| Claude 3.5 Sonnet | 54.0 | 27.5 | 29.6 | 43.7 | 58.3 | 20.3 | 17.8 | 34.2 |
| *Open-source MLLMs* | | | | | | | | |
| LLaVA-Med | 50.0 | 12.3 | 28.1 | 24.3 | 69.2 | 27.0 | 33.4 | 29.5 |
| LLaVA-1.5-7B | 50.0 | 12.3 | 28.1 | 27.2 | 63.9 | 19.2 | 25.3 | 25.2 |
| LLaVA-1.5-13B | 50.0 | 35.2 | 48.4 | 20.7 | 53.9 | 13.1 | 13.6 | 21.2 |
| LLaVA-1.6-Vicuna-7B | 50.0 | 15.8 | 29.4 | 16.5 | 50.1 | 1.0 | 13.6 | 13.3 |
| LLaVA-1.6-Vicuna-13B | 50.0 | 20.1 | 38.3 | 5.9 | 53.0 | 3.6 | 16.6 | 22.0 |
| LLaVA-1.6-34B | 50.2 | 19.9 | 36.0 | 17.0 | 57.2 | 12.8 | 16.6 | 22.4 |
| LLaVA-OneVision-7B | 49.8 | 11.4 | 34.5 | 30.0 | 58.7 | 17.0 | 20.6 | 20.4 |
| LLaVA-OneVision-72B | 50.6 | 29.6 | 50.4 | 40.6 | 52.3 | 7.0 | 13.1 | 25.0 |
| Deepseek-VL-Chat-7B | 50.9 | 15.7 | 27.9 | 15.6 | 63.7 | 27.5 | 22.4 | 21.1 |
| Idefics2-8B | 50.7 | 21.9 | 31.2 | 10.6 | 49.0 | 17.9 | 47.9 | 26.1 |
| Mantis-8B-siglip-Llama3 | 50.6 | 20.4 | 30.0 | 16.0 | 57.5 | 17.9 | 15.7 | 23.8 |
| MiniCPM-V-2.6 | 49.0 | 37.7 | 63.8 | 15.4 | 56.6 | 25.3 | 22.0 | 20.8 |
| Phi-3-Vision-128k-Instruct | 50.0 | 29.6 | 48.4 | 20.2 | 69.6 | 22.6 | 38.8 | 28.4 |
| Qwen2-VL-7B | 51.3 | 22.4 | 30.8 | 43.0 | 60.7 | 24.8 | 20.5 | 20.4 |
| Qwen2-VL-72B | 54.0 | 28.3 | 30.2 | 48.9 | 60.6 | 23.6 | 16.1 | 23.7 |
| InternVL2-8B | 50.6 | 14.3 | 27.8 | 38.1 | 55.8 | 16.1 | 17.7 | 22.3 |
| InternVL2-40B | 51.2 | 18.7 | 34.6 | 41.8 | 56.7 | 16.2 | 17.4 | 18.2 |
| InternVL2-Llama3-76B | 50.4 | 9.4 | 35.6 | 41.4 | 59.0 | 20.2 | 20.5 | 21.8 |
| PULSE-7B (Ours) | **82.4** | **74.8** | **11.0** | **61.3** | **90.7** | **85.4** | **5.0** | **73.8** |
| Δ over best proprietary MLLM | +27 | +47 | +15 | +11 | +30 | +61 | +10 | +39 |
| Δ over best open-source MLLM | +28 | +37 | +17 | +12 | +21 | +58 | +8 | +44 |

Table 3: In-domain evaluation results. Results marked as $^\dagger$ are copied from other papers. Results marked as $^*$ are obtained using the provided online software to collect prediction results. N/A: methods are not applicable or not designed for certain tasks. -: scores are not reported in the original papers. Note that the experimental setup of some domain-specific methods is not exactly the same as ours, thus the results listed are for reference purposes.

in terms of arena score. These results highlight PULSE's relative strength in handling complex, clinically-oriented ECG interpretation and analysis. Additionally, the performance gap across models on this challenging benchmark indicates considerable room for future improvements in this task.

## 4.4 ABLATION STUDY

**Effect of Training Data Source.** Given that ECGInstruct is compiled from diverse datasets, it is crucial to examine how each dataset contributes to the model's overall performance. Table 5 presents a comparative analysis of models trained on various dataset combinations. The model trained exclusively on PTB-XL (P) exhibits the lowest performance across all datasets, indicating the limitations of relying on a single data source for effective generalization. As we progressively incorporate additional datasets into the training set, the model's performance consistently improves. These results highlight the importance of curating diverse training data, as expanding beyond a single source enhances the model's capacity to generalize across datasets and tasks.

**Effect of Instruction Task.** To understand the individual contribution of each ECG-related task to model performance, we analyze combinations of four instruction tasks. As shown in Table 6, adding more tasks progressively improves performance across multiple benchmarks. Models trained solely

| Datasets | CPSC 2018 | | | CSN | G12EC | MMMU ECG | ECG Arena |
|---|---|---|---|---|---|---|---|
| Metric | AUC | F1 | HL | Accuracy | Accuracy | Accuracy | Arena Score |
| Random | 51.2 | 15.1 | 28.8 | 11.6 | 12.1 | 24.2 | 0 |
| Domain-specific Methods | | | | | | | |
| METS | - | - | - | N/A | N/A | N/A | N/A |
| MERL | **82.8**[†] | - | - | N/A | N/A | N/A | N/A |
| ST-MEM | 70.4[†] | - | - | N/A | N/A | N/A | N/A |
| ECG-GPT | 69.3* | 44.0* | 9.9* | N/A | N/A | N/A | N/A |
| Proprietary MLLMs | | | | | | | |
| GPT-4o | 50.9 | 10.6 | 18.2 | 57.5 | 49.2 | 43.5 | 33.5 |
| GPT-4o mini | 49.2 | 11.0 | 25.5 | 32.1 | 33.2 | 39.5 | 30.1 |
| Gemini-1.5-Pro | 50.1 | 7.4 | 20.5 | 50.5 | 36.0 | 40.0 | 31.2 |
| Claude 3.5 Sonnet | 52.8 | 11.5 | 18.9 | 51.5 | 51.4 | 42.0 | 37.1 |
| Open-source MLLMs | | | | | | | |
| LLaVA-Med | 50.0 | 2.5 | 20.2 | 13.8 | 14.1 | 27.0 | 15.9 |
| LLaVA-1.5-7B | 50.0 | 2.5 | 20.0 | 32.1 | 25.4 | 33.0 | 12.7 |
| LLaVA-1.5-13B | 50.4 | 13.3 | 30.1 | 30.7 | 30.7 | 35.0 | 13.1 |
| LLaVA-1.6-Vicuna-7B | 50.5 | 19.7 | 66.0 | 23.7 | 23.3 | 28.0 | 16.0 |
| LLaVA-1.6-Vicuna-13B | 50.0 | 19.3 | 62.8 | 31.4 | 35.0 | 38.0 | 17.9 |
| LLaVA-1.6-34B | 49.6 | 19.3 | 62.8 | 44.3 | 45.9 | 31.0 | 17.5 |
| LLaVA-OneVision-7B | 49.6 | 8.0 | 28.3 | 23.3 | 25.7 | 26.0 | 22.5 |
| LLaVA-OneVision-72B | 51.5 | 12.8 | 29.4 | 44.0 | 42.6 | 35.0 | 15.5 |
| Deepseek-VL-Chat-7B | 50.7 | 6.0 | 20.0 | 35.7 | 32.9 | 34.5 | 15.3 |
| Idefics2-8B | 49.0 | 17.9 | 47.9 | 22.8 | 26.2 | 36.0 | 4.9 |
| Mantis-8B-siglip-Llama3 | 51.3 | 19.1 | 48.5 | 17.6 | 22.6 | 38.5 | 13.6 |
| MiniCPM-2.6 | 50.0 | 18.0 | 48.4 | 12.7 | 19.6 | 34.5 | 20.4 |
| Phi-3-Vision-128k-Instruct | 50.6 | 19.0 | 70.2 | 14.8 | 18.4 | 31.0 | 11.3 |
| Qwen2-VL-7B | 49.4 | 17.5 | 46.3 | 25.5 | 32.9 | 31.5 | 8.5 |
| Qwen2-VL-72B | 50.7 | 9.8 | 18.9 | 35.5 | 42.9 | 35.0 | 10.3 |
| InternVL2-8B | 52.1 | 8.2 | 22.2 | 47.7 | 37.5 | 30.0 | 22.9 |
| InternVL2-40B | 52.4 | 8.2 | 21.4 | 41.0 | 45.0 | 30.5 | 28.0 |
| InternVL2-Llama3-76B | 51.3 | 6.5 | 20.4 | 26.6 | 34.7 | 38.0 | 22.5 |
| PULSE (Ours) | 76.9 | **57.6** | **8.6** | **85.2** | **78.2** | **58.0** | **38.9** |
| Δ over best proprietary MLLM | +24 | +46 | +10 | +28 | +27 | +15 | +2 |
| Δ over best open-source MLLM | +25 | +38 | +10 | +38 | +33 | +20 | +11 |

Table 4: Out-of-domain evaluation results. Results marked as [†] are copied from original papers. Results marked as * are obtained using the provided online software to collect prediction results. N/A: methods are not applicable or not designed for certain tasks. -: scores are not reported in the original papers.

| Training Data | PTB-XL Super | PTB-XL Report | CSN | CODE-15 | ECQ-QA | CPSC | G12 | MMMU ECG | ECG Arena | AVG |
|---|---|---|---|---|---|---|---|---|---|---|
| P | 68.2 | 56.7 | 82.8 | 31.5 | 31.8 | 23.4 | 65.4 | 40.0 | 28.4 | **-20.6** |
| P + M | 74.1 | 62.4 | 88.7 | 48.5 | 35.8 | 52.4 | 78.8 | 58.5 | 37.0 | **-8.6** |
| P + M + C | 74.1 | 63.8 | 87.5 | 85.8 | 43.4 | 51.0 | 75.5 | 55.5 | 39.4 | **-4.1** |
| P + M + C + E | 74.8 | 61.3 | 85.2 | 85.4 | 73.8 | 57.6 | 78.2 | 58.0 | 38.9 | **68.1** |

Table 5: Performance of different training dataset combinations. P: PTB-XL, M: MIMIC-IV-ECG, C: CODE-15%, E: ECG-QA. For PTB-XL Super, CODE-15% and CPSC, F1 scores are reported. For CSN, ECQ-QA, G12 and MMMU ECG, accuracy is reported. For PTB-XL Report, report scores are reported. For ECG Arena, arena scores are reported. **AVG** denotes the average across all metrics.

on basic feature recognition (F) performed poorly across all metrics, highlighting the limitations of a single-task approach. In contrast, the sequential addition of tasks led to substantial performance gains across multiple benchmarks. The model incorporating all four tasks achieved the highest performance, indicating a more comprehensive understanding of ECG images.

| Instruction Task | PTB-XL Super | PTB-XL Report | CSN | CODE-15 | ECQ-QA | CPSC | G12 | MMMU ECG | ECG Arena | AVG |
|---|---|---|---|---|---|---|---|---|---|---|
| F | 12.3 | 36.0 | 56.6 | 11.2 | 54.8 | 2.5 | 11.2 | 34.0 | 12.4 | **-42.5** |
| F + R | 26.9 | 54.0 | 83.8 | 73.3 | 61.4 | 31.0 | 67.3 | 47.5 | 25.3 | **-15.9** |
| F + R + M | 70.4 | 57.6 | 85.2 | 82.7 | 68.6 | 43.8 | 71.0 | 52.5 | 30.4 | **-5.7** |
| F + R + M + C | 74.8 | 61.3 | 85.2 | 85.4 | 73.8 | 57.6 | 78.2 | 58.0 | 38.9 | **68.1** |

Table 6: Performance of different ECG-related instruction task combinations. F: basic feature recognition, R: heart rhythm analysis, M: morphology and pathology identification, C: clinical report generation. For PTB-XL Super, CODE-15% and CPSC, F1 scores are reported. For CSN, ECQ-QA, G12, and MMMU ECG, accuracy is reported. For PTB-XL Report, report scores are reported. For ECG Arena, arena scores are reported. **AVG** denotes the average across all metrics.

### 4.5 Case Study

We further present some examples from our benchmark, comparing the outputs of our model with GPT-4o for ECG report generation (Appendix Figs. A11-A13) and ECG Arena (Appendix Fig. A14). While GPT-4o is capable of generating reports and answering questions by following instructions, it often produces responses that, although well-structured and seemingly relevant, contain significant inaccuracies in interpretation. In contrast, `PULSE` consistently provides more accurate responses that align closely with the ground truths. Additionally, we observed that GPT-4o tends to over-rely on its OCR capabilities when textual information is present in images, leading to superficial reasoning based on text rather than a deep analysis of visual data. For instance, in Appendix Fig. A13, GPT-4o identifies a left axis deviation based on the printed QRS axis degree in the image, without analyzing the visual waveform patterns. If such axis information were absent, the model would likely fail to identify the deviation.

### 4.6 Discussion

While the model demonstrates superior performance across various evaluation datasets, it faces notable challenges with more complex and open-ended tasks, such as report generation and multi-turn conversations. To further investigate the model's performance in report generation, we present the score breakdown in Fig. 4. The model excels in rhythm interpretation but struggles with waveform and diagnosis identification. These results suggest that future efforts should prioritize increasing the dataset's coverage of waveform and diagnosis-related cases to enhance the model's ability to detect these abnormalities. Additionally, as diagnosis identification may require more advanced multi-step reasoning, future research could focus on incorporating step-wise instruction tuning data to strengthen the model's reasoning capabilities. More discussion is provided in Appendix I.

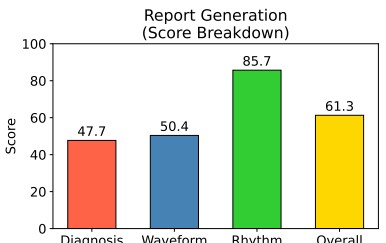

Figure 4: Score breakdown of report generation performance.

## 5 Conclusion

In this paper, we study the problem of ECG image interpretation, which is a crucial task in assessing cardiac conditions. We develop a new MLLM, `PULSE`, fine-tuned on the newly created `ECGInstruct` dataset with over 1 million samples across a diverse range of ECG-related tasks. Evaluated on the proposed benchmark, `ECGBench`, our model shows state-of-the-art performance, surpassing both proprietary and open-source MLLMs across multiple in-domain and out-of-domain evaluation datasets. This work demonstrates the potential of using MLLMs for enhancing ECG image analysis and interpretation in clinical applications.

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

## A    RELATED WORK

**Domain-specific Models for ECG.**    Many domain-specific models have been proposed to enhance automatic ECG diagnosis (Hannun et al., 2019; Ribeiro et al., 2020; Hughes et al., 2021). For example, Ribeiro et al. (2020) applied convolutional neural networks (CNNs) to encode ECG signals for diagnosing 6 types of abnormalities. To reduce dependence on high-quality labeled data, recent studies (Li et al., 2024c; Liu et al., 2024a; Na et al., 2023) have further explored self-supervised learning approaches using unlabeled ECG training data. For example, Liu et al. (2024a) proposed an ECG representation learning framework by integrating the ECG signals and clinical reports, showing improved performance in zero-shot ECG classification tasks. Despite these successes, most approaches treat ECG data as temporal physiological signals, which could be limiting in certain resource-constrained or remote settings where only printed or digital images are available. Recently, a few methods (Sangha et al., 2022; 2023; Khunte et al., 2024) have been proposed for ECG diagnosis using ECG images. For example, Khunte et al. (2024) developed a diagnostic report generation framework for ECG images, which is built upon a BEiT (Bao et al., 2021) vision transformer encoder and a GPT-2 (Radford et al., 2019) decoder. However, their model is only capable of the clinical report generation task, without generalizability to other diverse tasks. In contrast, our study investigates the capabilities of MLLMs for ECG image interpretation. We curate a diverse range of instruction tuning datasets to fine-tune the model, thus improving model generalizability.

**MLLMs in Healthcare**    Recent advancements in MLLMs have shown promising results in various healthcare domains. General medical multimodal models such as LLaVA-Med (Li et al., 2024a), MedPaLM (Singhal et al., 2023a;b), and Med-Gemini (Saab et al., 2024) have demonstrated capabilities in processing diverse medical data types. Additionally, domain-specific multimodal models have been developed for specialized fields like pathology (Lu et al., 2024b; Xu et al., 2024) and radiology (Wu et al., 2023). These models have shown potential in integrating visual and textual information to support clinical decision-making and medical education. However, despite the importance of ECG data in cardiac diagnosis and monitoring, current MLLMs often struggle to process ECG images effectively. This limitation highlights a significant gap in the application of MLLMs to cardiology, where the ability to interpret both visual ECG representations and accompanying clinical information is crucial.

**Multimodal Instruction Tuning.**    Instruction tuning has proven effective in the multimodal domain, particularly in vision-language models like LLaVA (Liu et al., 2024d), MiniGPT-4 (Zhu et al., 2023) and InstructBLIP (Dai et al., 2023). These models demonstrate impressive generalizability across various visual understanding and reasoning tasks. While multimodal instruction tuning has been applied to general medical imaging tasks (Li et al., 2024b; Singhal et al., 2023a), its application to ECG images remains largely unexplored. A recent work (Wan et al., 2024) introduced a targeted instruction tuning framework and fine-tuned existing open-source LLMs for ECG report generation. However, their approach is limited by a single-task instruction dataset focused solely on report generation, potentially constraining its adaptability to other ECG-related tasks. Moreover, their work also treats ECG data as temporal signals, whereas our paper focuses on encoding ECG images with MLLMs, which is more applicable to real scenarios where only printed or digital ECG images are available.

## B    PRELIMINARY ON 12-LEAD ECG

ECG is a vital diagnostic tool that measures the electrical activity of the heart over time, providing insights into both spatial and temporal aspects of cardiac function. Typically, an ECG recording is presented as a 12-lead multivariate time series, where each lead offers a unique perspective on heart activity. The six limb leads (I, II, III, aVR, aVL, and aVF) assess the electrical movements across the arms and legs, giving views from the frontal plane. Simultaneously, the six precordial leads (V1, V2, V3, V4, V5, and V6) monitor the chest, offering horizontal plane views. In this paper, we focus on ECG images that are synthesized from raw signals.

## C    DETAILS OF ECG IMAGE SYNTHESIS

We employ the ECG-image-kit (Shivashankara et al., 2024) framework to synthesize diverse ECG images from raw signal data. This toolkit allows for the generation of ECG images under various conditions by introducing a range of distortions and noises to better simulate real-world clinical data.

Specifically, in addition to generating standard 12-lead ECG images—characterized by black waveforms on a white background, red grid lines, and a 4x3 layout—we introduce a variety of perturbations to the images. These modifications include the addition of wrinkles and creases, simulating the physical wear and tear commonly observed in paper-printed ECGs. Our image synthesis process includes various augmentation methods to simulate physical distortions, image quality variations, and layout alterations. We introduce wrinkles and creases to mimic wear and tear commonly observed in paper-printed ECGs, and apply random rotations at varying angles to simulate misaligned scans or prints. To account for different acquisition systems and scanning qualities, we vary image resolutions and introduce random background colors, such as slight yellowing to represent aging or poor scanning quality. We also add noise to the images to simulate imperfections in the scanning or printing process. Furthermore, we experiment with different aspect ratios, overall image sizes, and ECG plot positions within the image to reflect the heterogeneity of ECG printouts across different systems and formats. In some cases (with a 0.02 probability), we randomly remove grid lines to account for variations in ECG presentation.

To further enrich the synthetic images, we randomly insert meta-information into the image header to simulate the annotations typically seen in clinical ECG reports. For the PTB-XL dataset, we extract patient demographics (e.g., age, gender) and basic ECG features (e.g., heart rate, axis deviations) from the associated PTB-XL feature annotation dataset, PTB-XL+ (Strodthoff et al., 2023). This extracted data is used to impute realistic meta-information, which is then randomly printed on the synthesized image. This random insertion of meta-data not only increases the visual variety of the images but also provides additional context, simulating real-world ECG prints that include patient and diagnostic information. To further increase diversity, we adopt alternative lead configurations beyond the standard 4x3 layout, such as 12x1 (single row of leads), 6x2 (two rows of six leads), and other commonly used clinical formats. These variations ensure that our model is exposed to a wide range of ECG presentation styles.

The augmentation process is designed to balance the dataset, with an approximate ratio of 1:1 between augmented and standard ECG images. This balance ensures that the model is exposed to both clean and distorted images, aiding in its generalization to real-world clinical scenarios.

# D  DETAILS OF INSTRUCTION TUNING DATASETS

| Source Dataset | Task | Type | # Samples |
|---|---|---|---|
| PTB-XL | Basic Feature Recognition | Close-ended QA | 22,759 |
| | | Open-ended QA | 906 |
| | | Fill-in-blank | 449 |
| | | Multi-choice QA | 5,716 |
| | Heart Rhythm Analysis | Close-ended QA | 19,550 |
| | | Open-ended QA | 201 |
| | | Fill-in-blank | 436 |
| | | Multi-choice QA | 16,179 |
| | Morphology and Pathology Identification | Close-ended QA | 50,166 |
| | | Open-ended QA | 2,649 |
| | | Fill-in-blank | 680 |
| | | Multi-choice QA | 13,432 |
| | Clinical Report | Open-ended QA | 16,302 |
| PTB-XL Total | | | 149,425 |
| ECG-QA | Basic Feature Recognition | Close-ended QA | 40,154 |
| | Heart Rhythm Analysis | Close-ended QA | 8,993 |
| | Morphology and Pathology Identification | Close-ended QA | 90,211 |
| ECG-QA Total | | | 139,358 |
| MIMIC-ECG | Basic Feature Recognition | Close-ended QA | 759 |
| | | Open-ended QA | 4,759 |
| | | Fill-in-blank | 6,470 |
| | | Multi-choice QA | 17,186 |
| | Heart Rhythm Analysis | Close-ended QA | 48,625 |
| | | Open-ended QA | 5,262 |
| | | Fill-in-blank | 11,487 |
| | | Multi-choice QA | 49,352 |
| | Morphology and Pathology Identification | Close-ended QA | 8,241 |
| | | Open-ended QA | 81,080 |
| | | Fill-in-blank | 18,264 |
| | | Multi-choice QA | 61,456 |
| | Clinical | Open-ended QA | 486,746 |
| MIMIC-ECG Total | | | 799,687 |
| CODE-15% | Basic Feature Recognition | Close-ended QA | 22,177 |
| | Heart Rhythm Analysis | Close-ended QA | 13,893 |
| | Morphology and Pathology Identification | Close-ended QA | 31,570 |
| CODE-15% Total | | | 67,640 |
| `ECGInstruct` Total | | | 1,156,110 |

Table A1: Detailed data statistics of `ECGInstruct`.

---

**Training Data Examples: Basic Feature Recognition**

**Instruction:**
What are the main ECG features I'm seeing here?
**Response:**
Let's break it down. We have a rhythm that appears to be irregular, with varying P-P intervals. The heart rate is not explicitly stated, but based on the ECG tracing, it seems to be within the normal range. The cardiac axis appears to be normal as well. One notable feature is the presence of an rSr' pattern in lead V1, which is likely a normal variant. Additionally, we can see some nonspecific T wave changes in the septal leads.
**Follow-up Instruction:**
Can you explain the significance of the rSr' pattern in lead V1? Is it a cause for concern?
**Response:**
The rSr' pattern in lead V1 is often seen in healthy individuals, especially in younger people. It's considered a normal variant because it doesn't necessarily indicate any underlying cardiac pathology. The "r" represents a small initial positive deflection, followed by a larger negative deflection (the "S" wave), and then a secondary positive deflection (the "r'"). This pattern is thought to be related to the normal activation sequence of the right ventricle. Since it's isolated to lead V1 and doesn't appear to be associated with any other abnormalities, it's unlikely to be a cause for concern in this case.

**Sub-task:** Basic Waveform Understanding; **Question type:** Open-ended; **Source dataset:** MIMIC-IV-ECG

**Instruction:**
What can be inferred about the heart's electrical conduction system based on this ECG?
Options:
A. There is a block in the electrical conduction pathway
B. There is an abnormality in the heart's electrical conduction system
C. The heart's electrical conduction system is functioning normally
D. There is a rhythm disturbance
**Response:**
This image indicates that the heart's electrical conduction system is functioning normally, as there are no apparent abnormalities or disturbances in the electrical activity. The presence of a sinus rhythm and a normal axis suggests that the heart's electrical conduction system is operating within normal limits, without any significant defects or anomalies. This ECG image is consistent with a normal heart, and there is no evidence of any conduction system abnormalities.
We can conclude that the answer is C. The heart's electrical conduction system is functioning normally

**Sub-task:** Conduction System Interpretation; **Question type:** Multi-choice QA; **Source dataset:** PTB-XL

**Instruction:**
You are reviewing an ECG tracing in a clinical setting. Based on the ECG morphology, fill in the blanks:
This ECG tracing is most consistent with a rhythm of ___________________, with a cardiac axis of _________________, and a bundle branch block of the __________ bundle.
**Response:**
This ECG tracing is most consistent with a rhythm of **atrial fibrillation**, with a cardiac axis of **left axis deviation**, and a bundle branch block of the **left** bundle.

**Sub-task:** Axis and Rhythm Determination; **Question type:** Fill-in-the-blank; **Source dataset:** MIMIC-IV-ECG

**Instruction:**
Given this ECG image, please answer the following question. From the provided options, select all that apply. List your selections, separated by slashes.
Question:
Which ECG leads are affected by baseline drift?
Options:
llead III, lead V2, none, lead V4, lead aVL, lead V5, lead V6, lead aVF, lead V3, lead aVR, lead V1, lead I, lead II. Provide the accurate response without additional details.
**Response:**
lead aVF/lead V3

**Sub-task:** Lead Placement Determination; **Question type:** Close-ended QA; **Source dataset:** ECG-QA

Figure A1: The Examples of basic feature recognition instructions for finetuning PULSE.

---

**Training Data Examples: Heart Rhythm Analysis**

**Instruction:**
Analyze the ECG image and identify the combination of cardiac rhythm abnormalities that can be diagnosed simultaneously in this patient. Rank the abnormalities in order of their potential impact on the patient's hemodynamic stability, with 1 being the most critical. Use a semicolon-separated list with the rhythm abnormality followed by its severity ranking

**Response:**
Sinus tachycardia; 1; Right bundle branch block; 3; Premature atrial contraction; 2; Premature ventricular contraction; 2.
Justification: Sinus tachycardia is ranked as the most critical abnormality since it can lead to decreased cardiac output, increased myocardial oxygen demand, and potentially exacerbate underlying cardiac conditions. Premature atrial contractions and premature ventricular contractions are ranked equally as they can both disrupt normal cardiac function, although their individual impact might be less severe than sinus tachycardia. Right bundle branch block, while still a significant abnormality, is ranked last as it may not have an immediate impact on hemodynamic stability, but rather serves as an indicator of underlying cardiac pathology.

**Sub-task:** Rhythm Analysis; **Question Type:** Open-ended; **Source Dataset:** PTB-XL

**Instruction:**
Imagine you're an ER doctor reviewing this ECG, what would be your primary concern regarding the patient's heart rhythm? Directly output one of the following options without explanation.
**Options:**
A) Tachycardia; B) Ventricular Fibrillation; C) Atrial Fibrillation; D) Bradycardia;
E) Long QT Syndrome; F) First-Degree AV Block; G) Second-Degree AV Block;
H) Normal Sinus Rhythm
**Response:**
D) Bradycardia

**Sub-task:** Arrhythmia Identification; **Question Type:** Multi-choice QA; **Source Dataset:** MIMIC-IV-ECG

**Instruction:**
If you were to categorize the ECG rhythm of this patient into one of the following options:
Normal Sinus Rhythm, Atrial Fibrillation, Ventricular Tachycardia, or Sinus Rhythm with Premature Atrial Contractions,
which one would you choose? ________________
**Response:**
Sinus Rhythm with Premature Atrial Contractions

**Sub-task:** Rhythm Classification; **Question Type:** Fill-in-the-blank; **Source Dataset:** PTB-XL

**Instruction:**
As part of a heart health assessment, your task is to examine the ECG, identify all fitting options from the provided list, and enumerate them using semicolon:
1dAVb(1st degree av block), RBBB(right bundle branch block), LBBB(left bundle branch block), SB(sinus bradycardia), ST(sinus tachycardia), AF(atrial fibrillation).
Only answer based on the given options without any explanation.
**Response:**
RBBB(right bundle branch block)

**Sub-task:** Conduction Abnormality Detection; **Question Type:** Close-ended QA; **Source Dataset:** CODE-15%

Figure A2: The Examples of heart rhythm analysis instructions for finetuning PULSE.

---

### Training Data Examples: Morphology and Pathological Condition Identification

**Instruction:**
What specific features of the ECG tracing suggest the presence of a right bundle branch block, and how do these features relate to the overall cardiac conduction pattern, considering the morphology of the QRS complexes in leads V1 and V2? Provide a detailed explanation in 2-3 sentences.

**Response:**
The ECG tracing exhibits a right bundle branch block (RBBB) due to the characteristic wide S waves in leads I and V6, accompanied by QS complexes in leads V2, indicative of delayed right ventricular depolarization. The QRS complexes in lead V1 display a prolonged duration, exceeding 140 ms, with a characteristic "M" or "W" shape, further supporting the diagnosis of RBBB. This conduction pattern is a result of a block in the right bundle branch, leading to an altered sequence of ventricular activation, causing the delayed and abnormal depolarization of the right ventricle.

**Sub-task:** Wave Morphology Analysis; **Question Type:** Open-ended; **Source Dataset:** PTB-XL

---

**Instruction:**
Answer the following multiple choice question. Think step by step before answering.
What is the likely location of a previous myocardial infarction (MI) based on this ECG?
**Options:**
A. Anterior wall    B. Lateral wall    C. Inferior wall    D. Septal wall

**Response:**
The ECG shows a pattern consistent with a previous MI, as evidenced by the abnormal Q-waves and ST-segment changes. The location of the MI can be inferred by the leads involved. We can observe that the inferior leads (II, III, and aVF) show the most prominent changes, indicating that the MI likely occurred in the inferior wall. Based on this, we select C. Inferior wall

**Sub-task:** Interpretation of Ischemia or Infarction; **Question Type:** Multi-choice QA; **Source Dataset:** MIMIC-IV-ECG

---

**Instruction:**
Compare the QRS complex morphology in leads V1-V4 to that of a normal sinus rhythm ECG. Explain the significance of the observed differences, considering the context of intraventricular conduction disorder and delayed R progression.
Then, fill in the blanks: The QRS complex morphology in leads V1-V4 is most indicative of _________. (Provide a detailed explanation in 2-3 sentences before filling in the blank.)

**Response:**
The QRS complex morphology in leads V1-V4 exhibits a notable difference from that of a normal sinus rhythm ECG. Specifically, the delayed R progression and intraventricular conduction disorder in V4 suggest a disruption in the normal ventricular depolarization sequence, resulting in altered QRS complex morphology. This deviation is significant because it indicates a pathological EKG pattern, which may be associated with underlying cardiac conditions such as left ventricular hypertrophy. The QRS complex morphology in leads V1-V4 is most indicative of **pathological left ventricular conduction**.

**Sub-task:** Pathological Conduction Identification; **Question Type:** Fill-in-the-blank; **Source Dataset:** PTB-XL

---

**Instruction:**
Imagine you are an expert cardiologist reviewing this ECG image. Considering the patient's history and this ECG, what could have contributed to the development of the prolonged QT interval?
Rank your top three possibilities in order of likelihood, using a semicolon-separated list.

**Response:**
Hypokalemia; certain medications or drugs; genetic mutations or channelopathies.

**Sub-task:** Electrolyte imbalances Interpretation; **Question Type:** Open-ended QA; **Source Dataset:** PTB-XL

---

Figure A3: The Examples of morphology and pathological condition identification instructions for finetuning `PULSE`.

**Training Data Examples: Clinical Reporting**

**Instruction:**
Review this electrocardiogram image and provide your expert diagnosis.

**Response:**
The electrocardiogram (ECG) reveals a sinus tachycardia, characterized by a heart rate exceeding 100 beats per minute, indicating an abnormal acceleration of the heart rhythm. This is further accentuated by the presence of an exaggerated left type, suggesting an increased electrical activity in the left ventricle.
Additionally, peripheral hypotension is noted, indicating lower-than-normal blood pressure in the peripheral arteries, which may be a concern as it can lead to inadequate perfusion of vital organs.The ECG also demonstrates an incomplete right bundle branch block (RBBB), which suggests a delay or obstruction in the electrical conduction pathway of the right ventricle.
Furthermore, the QRS complex is abnormal, with an altered morphology and duration, indicating a disruption in the ventricular depolarization process. The presence of an old inferior infarct is likely, suggesting a previous myocardial infarction (MI) in the inferior wall of the heart.
Moreover, the ECG shows signs of old ST and T wave abnormalities, indicating possible previous episodes of ischemia or injury to the myocardium. The ECG also suggests likely anterior ischemia or left ventricular strain, which may indicate ongoing or previous damage to the anterior wall of the heart.

**Question Type:** Open-ended; **Source Dataset:** PTB-XL

**Instruction:**
Examine the ECG signal image and produce a thorough report based on your findings, ending with the final diagnosis.

**Response:**
The electrocardiogram (ECG) reveals atrial fibrillation with a rapid ventricular response, indicating a chaotic and rapid heart rhythm originating from the atria. Additionally, there are signs of a possible anterior myocardial infarction, although the age of the infarct cannot be determined from this ECG. Furthermore, ST-T changes in the inferior and lateral leads suggest myocardial ischemia, which may be partly attributed to the rapid heart rate and rhythm, but also raises concerns for underlying coronary artery disease.
Final Diagnosis: Atrial fibrillation with rapid ventricular response, possible anterior myocardial infarction, and signs of myocardial ischemia.

**Question Type:** Open-ended; **Source Dataset:** MIMIC-ECG

Figure A4: The Examples of clinical reporting instructions for finetuning PULSE.

## E   PROMPTS

---

**Prompt: Multi-task Data Synthesizing**

---

Your task: Create a complex ECG visual task based on the given report and target task type:

Guidelines for task creation:
1. Design a concise yet challenging graduate-level task that requires deep reasoning.
2. Frame the task as interacting with an actual ECG image, without mentioning the report. Make the task visually centric, assuming direct ECG image analysis.
3. Strictly base all information on the given ECG report only. Avoid tasks and answers that are inconsistent with the report.
4. Avoid restating the report or using phrases like "As described in the report."
5. Generate one task from a diverse range of task types, including but not limited to:
   Direct questions (e.g. "What is the heart rhythm?")
   Hypothetical scenarios (e.g. "Imagine you're an ER doctor reviewing this ECG...")
   Comparative tasks (e.g. "How does this ECG differ from a normal sinus rhythm?")
   Explanation requests (e.g. "Explain the significance of the QS complexes seen in V2.")
   Problem-solving scenarios (e.g. "Given these ECG findings, what further tests might you order?")
   Educational prompts (e.g. "Teach a medical student about the key features of this ECG.")
   Role-playing scenarios (e.g. "You're consulting with a cardiologist about this ECG. What do you tell them?")
   Decision-making tasks (e.g. "Based on this ECG, would you clear this patient for surgery? Why or why not?")
6. Specify a clear, appropriate output format within the task instructions(free-form, "think-step-by-step", direct output the short answer(in one phrase or one sentence), JSON format, table, list, different delimiters(such as commas, semicolons, numeric order), etc.). Do not limited to the given task type and format, you have the freedom to design any type of task you deem appropriate.
7. Focus the task on one or more of the following ECG analysis aspects:
   a. Basic ECG feature interpretation (e.g. heart rate, rhythm, cardiac axis)
   b. Diagnosis and classification (e.g. diagnosis identification, waveform classification, rhythm classification)
   c. Waveform and interval analysis (e.g. P wave morphology, PR interval, QT interval, QRS complexes, T wave morphology)
8. Ensure the task complexity aligns with the given report's information.

After creating the task:
1. Provide a detailed, accurate answer to your own task.
2. Ensure your answer is comprehensive and strictly based on the report.
3. Strictly follow the output format and requirements specified in your task instructions.

ECG Report:
{report}

Target Task Type:
{target}

Present your work in this format:
Task: [Concise content of the ECG tasks, including required output format. Do not include phrases like "Output format:..." or  like "[Insert image here]", but in more natural expression. ]

Response: [Comprehensive response following the task's requirements, strictly based on the report]

Do not include any content outside of the Task and Response sections.

---

Figure A5: The prompt used to synthesize ECG instruction tasks based on clinical reports.

---

**Prompt: Multi-turn Dialogue Synthesizing**

Your task: Create a 2-4 turn dialogue between a medical professional and an AI assistant analyzing an ECG, based on the given report:

Guidelines for dialogue creation:
1. Design a series of questions and answers that progressively explore the ECG findings in depth, suitable for graduate-level medical professionals.
2. Frame the dialogue as if the medical professional is directly analyzing an actual ECG image, without mentioning the report. Make the conversation visually centric, assuming direct ECG image analysis.
3. Strictly base all information on the given ECG report only. Avoid including details inconsistent with the report.
4. Do not use phrases like "As described in the report," "The report mentions," or "The term..." The dialogue should not appear to reference an external report.
5. Begin with direct questions about basic ECG features, then progress to more complex interpretations and clinical implications.
6. Include a mix of question types, with an emphasis on direct questions:
  - Direct questions (e.g., "What are the main ECG features?", "What is the heart rhythm?")
  - Requests for explanations (e.g., "Can you explain the significance of the QS complexes?", "What the cause of these features?")
  - Clinical reasoning questions (e.g., "Given these findings, what's your diagnosis?")
  - Hypothetical scenarios (e.g., "How would you manage a patient presenting with this ECG?")
7. Focus the dialogue on one or more of the following ECG analysis aspects:
  a. Basic ECG feature interpretation (e.g., heart rate, rhythm, cardiac axis)
  b. Diagnosis and classification (e.g. diagnosis identification, waveform classification, rhythm classification)
  c. Waveform and interval analysis (e.g. P wave morphology, PR interval, QT interval, QRS complexes, T wave morphology)
  d. Clinical implications and management
8. Ensure the dialogue complexity aligns with the given report's information.

After creating the dialogue:
1. Provide extremely comprehensive and detailed answers from the AI assistant's perspective. Each response should thoroughly cover all relevant aspects of the question asked.
2. Ensure all answers are comprehensive and strictly based on the report, without explicitly referencing it.
3. Make the dialogue flow naturally, as if a real user is progressively exploring the ECG findings.
4. Structure the AI assistant's responses to be highly readable:
  - Break down complex information into digestible parts.
  - Use bullet points or numbered lists to organize information
  - Include brief explanations of medical terms or concepts when necessary
  - Provide context for why certain findings are significant
 Aim for a balance between depth of information and clarity of presentation in each response.

ECG Report:
{report}

Present your work in this format:
Human: [First question about the ECG]

Assistant: [Comprehensive response based strictly on the report]

Human: [Follow-up question delving deeper into the ECG analysis]

Assistant: [Detailed answer providing further insights]

[Continue the dialogue for up to 2 more turns if necessary, ensuring a natural progression of inquiry]

Do not include any content outside of the dialogue format. Ensure that the entire conversation appears to be about analyzing an actual ECG image, without any indication that the information comes from a written report.

---

Figure A6: The prompt used to synthesize ECG multi-turn dialogue as instruction tuning data.

---

**Prompt: Report Revision**

---

I will provide you with an ECG report. Please expand the report into a comprehensive and detailed version, considering all aspects mentioned in the original report. The expanded version should be at least 4 sentences long. Ensure that you elaborate on each point from the original report, providing more context and explanation where possible. Do not add any new content, interpretations, or conclusions beyond what is explicitly stated in the original report. Avoid using phrases like "Here is the revised report" or similar introductions. Simply begin with the expanded content.

Original Report:
{report}
Expanded Report:

Figure A7: The prompt used to revise (and translate) original reports.

---

**Prompt: Instruction Data Scoring**

---

Task: Given an ECG report and a corresponding question-answer pair, score the quality of the answer based on the guidelines provided. The score should range from 0 to 5, where 0 represents poor quality and 5 represents excellent quality. You should be strict when giving the final assessment if some of the criteria are not satisfied. Please consider the following criteria for scoring:

**1. Relevance:** Does the answer directly address the question asked?
**2. Accuracy:** Is the information in the answer accurate and consistent with the ECG report?
**3. Usefulness:** Does the answer provide helpful information that would aid understanding or decision-making based on the ECG report?
**4. Constructed Information:** Does the answer invent details not present in the ECG report?
**5. Presence of Direct Report Quotation:** A good answer does not simply quote or directly replicate phrases from the ECG report. It should assume that the questioner does not know the report's specific content. The presence of direct report quotations is not allowed in the answer, otherwise, the overall scores should be at most 2.

Output format:
Please first output a single line containing a comprehensive explanation of your evaluation, avoiding any potential bias. In the subsequent line, please provide the value indicating the scores in the format: "Score: [your rating score]"

Please apply the above scoring guide to the following ECG report and question-answer pair:

ECG Report: {report}
Question: {question}
Answer: {answer}

Figure A8: The prompt used to score and filter generated instruction data.

---

**Prompt: Evaluation of Report Generation**

---

Evaluate the alignment and quality of a generated ECG report by comparing it to a ground truth clinician's report. The evaluation will focus on three key aspects: Diagnosis, Waveform, and Rhythm. Use specific criteria for each aspect and be precise in comparing medical terminologies. Only focus on information present in the ground truth report, identifying any mistakes. Remain objective and do not let the response length affect your evaluation.

Evaluation Criteria:
**1. Diagnosis (0-10):**
Assess how well the generated ECG report matches the clinical diagnoses in the ground truth report. Focus on conditions like conduction disturbances, ischemia, hypertrophy, and other abnormalities as presented in the ground truth report.
- 10: All key diagnoses are correctly identified with no errors or omissions.
- 5: Partially accurate, with some diagnoses identified correctly but key conditions missing or incorrect.
- 0: Fails to identify key diagnoses, with multiple critical errors.

**2. Waveform (0-10):**
Evaluate the accuracy and quality of the ECG waveform morphology in the generated report compared to the ground truth. Focus on abnormalities in P-wave, QRS complex, ST changes, T-wave, and intervals (PR, QT), ensuring waveform morphology is consistent with the ground truth.
- 10: All waveform abnormalities are correctly identified without errors.
- 5: Some waveform abnormalities are identified, but key issues are missed or misinterpreted.
- 0: Fails to identify key waveform abnormalities, with multiple critical errors.

**3. Rhythm (0-10):**
Assess the accuracy and clarity of rhythm interpretation in the generated report. Focus on identifying and describing normal and abnormal rhythms (e.g., sinus rhythm, atrial fibrillation, ventricular tachycardia) as presented in the ground truth report.
- 10: Rhythm interpretation is fully accurate and clearly described.
- 5: Rhythm interpretation is partially accurate but contains notable errors or omissions.
- 0: Rhythm interpretation is largely incorrect, with critical errors.

Please organize your output in a JSON format of diagnosis, form and rhythm, with a brief explanation of each aspect. For example: {Diagnosis: {Score: $SCORE$, Explanation: $EXPLANATION$}}

[The Start of Ground Truth Report]
{ground_truth_report}
[The End of Ground Truth Report]
[The Start of Generated Report]
{generated_report}
[The End of Generated Report]

Figure A9: The prompt used to evaluate the generated report.

---

**Prompt: Evaluation of ECG Arena**

---

Evaluate the quality of a model's response to an ECG-related question by comparing it with a given ground truth answer. Focus on three aspects: accuracy, completeness, and instruction adherence. Be precise and objective, especially when identifying errors in medical terminology. Do not let the response length affect your evaluation.

Evaluation Criteria:
**1. Accuracy (0-10):**
How well does the model's response match the ground truth, particularly in ECG interpretation and diagnosis? This score emphasizes whether the key information is correct, such as the correct identification of waveforms, intervals, and clinical diagnoses.
- 10: Fully accurate, with correct ECG interpretation, terminology, and diagnosis.
- 5: Partially accurate, with some correct information but notable errors or omissions.
- 0: Largely inaccurate or misleading.

**2. Completeness (0-10):**
Does the response cover essential aspects of ECG interpretation (e.g., rhythm, axis, waveforms, clinical causes) mentioned in the ground truth? This score focuses on whether the answer is comprehensive and includes as much essential information as possible.
- 10: Comprehensive, covering all key details.
- 5: Partially complete, with important points missing.
- 0: Incomplete, lacking critical information.

**3. Instruction Adherence (0-10):**
Does the model follow the specific instructions in the question (e.g., listing features, suggesting a diagnosis)? This score focuses on how well the model follows the task instructions, regardless of the correctness of the answer.
- 10: Fully follows instructions.
- 5: Partially follows instructions, with some deviations.
- 0: Fails to follow instructions or provides an irrelevant response.

Please organize your output in a JSON format of accuracy, completeness, and instruction adherence, with a brief explanation of each aspect. For example: {Accuracy: {Score: $SCORE$, Explanation: $EXPLANATION$}}

[The Start of Ground Truth Answer]
{ground_truth_answer}
[The End of Ground Truth Answer]

[The Start of Model's Response]
{model_response}
[The End of Model's Response

---

Figure A10: The prompt used to evaluate the ECG Arena.

# F  EXPERIMENTAL SETUP

## F.1  DETAILS OF EVALUATION METRICS

**Abnormality Detection.** we utilize multi-label classification metrics, including Macro AUC, Macro F1, and Hamming Loss, to evaluate the datasets PTB-XL Super, CODE-15%, and CPSC 2018, where multiple correct labels may exist. For the ECG-QA, CSN, and G12EC datasets, we adopt accuracy as the evaluation metric.

**Report Generation.** Rather than relying on traditional text generation metrics, we leverage strong LLMs as evaluators, following the approach of Zheng et al. (2024). This method provides a more nuanced evaluation by focusing on key aspects of the reports. Specifically, we use GPT-4o to compare the model-generated reports against those written by cardiologists. We introduce a "Report Perfect Score", which is based on three critical components of a generated report: (1) Rhythms (0 to 10 points), (2) Waveform Morphology (0 to 10 points), and (3) Diagnosis (0 to 10 points). The final score is the average of these three components, scaled to a maximum of 100 points. The prompt used to query GPT-4o for evaluating the report score is provided in Appendix Fig. A9.

**MMMU ECG.** We adopt accuracy as the primary metric. We have designed systematic, rule-based evaluation pipelines to ensure robust and consistent scoring. To mitigate the potential influence of any intermediate generations (e.g., reasoning steps) in long responses, we employ robust regular expressions and develop response-processing workflows. These are used to extract answer options from the long responses for accurate answer matching. In cases where no valid answer can be extracted from the model's response, we perform random selection to assign a score.

**ECG Arena.** We also employ a strong judge model, GPT-4o, to assess model performance by comparing generated responses with ground truth answers. The evaluation considers three perspectives, each scored on a scale of 0-10: Accuracy (how closely the model's response matches the ground truth), Completeness (whether the model provides a comprehensive answer covering all aspects of ECG interpretation), and Instruction Adherence (how well the model follows the specific instructions in the question). We calculate the final score by averaging these three aspects and scaling to a maximum of 100 points. The specific prompt used for GPT-4 evaluation is provided in Appendix Fig. A10.

## F.2  IMPLEMENTATION DETAILS

We follow the model architecture of LLaVA, which includes three core components: a vision encoder, a large language model, and a projector to align image and text modalities. Table A2 summarizes all the model parameters. Specifically, for the LLM, we utilize Vicuna-1.5-7B, while the vision encoder is based on CLIP-ViT-Large-Patch14-336. We employ a 2-layer MLP as a projector to map the visual features from the CLIP encoder onto the tokens used by the LLM. These features are mapped onto predefined image tokens, which encapsulate the features of ECG images. The tokens representing ECG features are then concatenated as an image context preceding the dialogue.

We format all datasets into a chatbot-style multi-turn dialogue format (same as Vicuna-1.5-7B) and use the special token <image> to represent image features within the text data. For example, a sample data instance is: "Human: <image> Describe this ECG image. \n Assistant: This image . . . ". To enhance the model's ability to handle ECG images of various sizes encountered in real-world scenarios, we employ Anyres. Anyres divides high-resolution images into multiple sub-images of size 336x336. The features of these sub-images are then concatenated with the global features of the original image to form the final image representation.

We fine-tune all parameters of the vision encoder, projector, and LLM. The training process uses a learning rate of 2e-5, a batch size of 128, and a cosine scheduler with a 5% warm-up period over three epochs. The loss is calculated using the cross-entropy loss function, focusing on the response portion of the dialogue.

| Model Parameters | |
| --- | --- |
| Total | 7.06B |
| Vision Encoder(clip-vit-large-patch14-336) | 303.5M |
| Connector | 21M |
| LLM(Vicuna-1.5-7B) | 6.74B |
| **Training Parameters** | |
| Learning Rate | 2e-5 |
| Weight Decay | 0.0 |
| Warmup Ratio | 0.03 |
| Learning Rate Scheduler | Cosine |
| Batch Size | 128 |
| **Vision Encoder Arch** | |
| Hidden Size | 1024 |
| Input Resolution | 336 |
| ViT Layer | 24 |
| ViT Heads | 16 |
| Patch Size | 14 |
| **LLM Arch** | |
| Hidden Size | 4096 |
| Max Context Length | 4096 |
| Attention Heads | 32 |
| Hidden Layers | 32 |
| KV Heads | 32 |

Table A2: Summary of model parameters.

# G    ADDITIONAL RESULTS

**Ablation study of ViT model.**    We conducted an ablation study by unfreezing the ViT module parameters during training and reported the model performance in Table A3 below. The results show a further performance improvement (i.e., average score from 71.8 to 75.0) compared to the original model with frozen ViT parameters.

| Models | PTB-XL Super | PTB-XL Report | CSN | CODE-15 | ECQ-QA | CPSC | G12 | MMMU ECG | ECG Arena | **AVG** |
|---|---|---|---|---|---|---|---|---|---|---|
| Unfrozen ViT | **76.9** | **65.4** | **87.9** | **87.0** | 71.6 | **65.4** | **81.4** | **64.0** | **41.5** | **75.0** |
| Frozen ViT | 74.8 | 61.3 | 85.2 | 85.4 | **73.8** | 57.6 | 78.2 | 58.0 | 38.9 | 71.8 |

Table A3: Ablation results on the impact of unfreezing ViT module parameters

# H  CASE STUDY

---

**Case Study: PTB-XL Report**

---

**Question:**
Please write a clinical report based on this ECG image.

**Ground Truth Report:**
Sinus rhythm. Prolonged PR interval. R-S transition zone in V leads displaced to the left. Voltages are high in limb leads suggesting left ventricular hypertrophy (LVH). ST segments are depressed in leads I, II, aVL, V5, and V6. T waves are flat in these leads. Non-specific but consistent with myocardial changes.

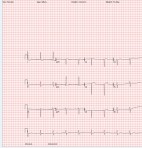

---

**PULSE Response:** The electrocardiogram (ECG) reveals a **sinus rhythm**, indicating that the heart is beating in a regular, rhythmic pattern, with a normal heart rate and P-wave morphology. **The PR interval is prolonged**, measuring longer than the normal range of 120-200 milliseconds, suggesting a delay in the conduction of electrical impulses from the atria to the ventricles. This prolongation may be indicative of a first-degree atrioventricular (AV) block, which can be a benign finding but may also be associated with underlying cardiac conditions. The ECG also demonstrates **high voltages in the limb leads**, specifically leads I, II, and III, which is **suggestive of left ventricular hypertrophy (LVH)**. This finding is consistent with an increase in the thickness of the left ventricular wall, which can be a response to chronic pressure overload or volume overload. **The ST segments are depressed in leads I, II, aVL, V5, and V6**, indicating a possible alteration in the repolarization pattern of the ventricles. This depression may be indicative of left ventricular strain or ischemia, which can be a sign of myocardial injury or infarction. Furthermore, the **T waves are low or flat in leads I, II, aVL, V5, and V6**, which may be a manifestation of left ventricular strain or ischemia. The combination of these findings, including the prolonged PR interval, high voltages in the limb leads, and ST segment depression, **suggests that the patient may be experiencing myocardial ischemia**, which is a condition where the heart muscle does not receive sufficient blood flow, leading to a mismatch between oxygen demand and supply. In conclusion, based on the ECG findings, the diagnosis is consistent with **myocardial ischemia, left ventricular hypertrophy, T wave abnormality, prolonged PR interval, and sinus rhythm**.

---

**LLM-as-Judge Feedback for PULSE Response:**
1. Diagnosis
**Score: 10.** Explanation: The generated report correctly identifies all key diagnoses present in the ground truth report, including sinus rhythm, prolonged PR interval, left ventricular hypertrophy (LVH), ST segment depression, and T wave abnormalities. The interpretation of these findings as consistent with myocardial ischemia is also accurate.
2. Form
**Score: 10.** Explanation: The generated report accurately describes the waveform abnormalities, including the prolonged PR interval, high voltages in the limb leads, ST segment depression in leads I, II, aVL, V5, and V6, and flat T waves in these leads. These descriptions match the ground truth report precisely.
3. Rhythm
**Score: 10.** Explanation: The generated report correctly identifies the rhythm as sinus rhythm, which is consistent with the ground truth report. The description of the rhythm is clear and accurate.
**Final Score: (10+10+10)/3 * 10 = 100**

---

**GPT-4o Response:**
Clinical Report: Electrocardiogram (ECG)
**ECG Interpretation:** Heart Rate: The heart rate appears to be within normal limits. Precise calculation of the heart rate is recommended for confirmation. Rhythm: The rhythm appears to be regular. There are consistent R-R intervals, suggesting a **regular sinus rhythm**.
P Waves: P waves are present before each QRS complex, indicating sinus rhythm. The morphology of the P waves appears normal. **PR Interval: The PR interval appears to be within normal limits (120-200 ms).** QRS Complex: The QRS duration appears to be within normal limits (less than 120 ms). There is no evidence of significant QRS complex abnormalities such as bundle branch block.
ST Segment: **The ST segments appear to be isoelectric (flat) in most leads, with no significant elevation or depression noted. T Waves: T waves appear to be upright in leads I, II, V3-V6, and inverted in aVR, which is normal. There are no significant T wave abnormalities noted.** Axis: The QRS axis appears to be normal. The exact axis calculation is recommended for confirmation.
Additional Observations: There are **no significant signs of ischemia, infarction, or other acute changes. No evidence of left ventricular hypertrophy or other chamber enlargement**.
**Conclusion:** This ECG shows a normal sinus rhythm **with no significant abnormalities. The patient's ECG is within normal limits for her age and clinical presentation.**

---

**LLM-as-Judge Feedback for GPT-4o Response:**
1. Diagnosis
**Score: 0.** Explanation: The generated report fails to identify key diagnoses present in the ground truth report. It does not mention the prolonged PR interval, left ventricular hypertrophy (LVH), ST segment depression, or flat T waves. Instead, it incorrectly states that there are no significant signs of ischemia, infarction, or other acute changes, and no evidence of left ventricular hypertrophy.
2. Form
**Score: 0.** Explanation: The generated report inaccurately describes the waveform morphology. It incorrectly states that the PR interval is within normal limits, fails to identify the high voltages in limb leads, and does not mention the ST segment depression or flat T waves. These are critical waveform abnormalities that are present in the ground truth report.
3. Rhythm
**Score: 5.** Explanation: The generated report correctly identifies a regular sinus rhythm, which is consistent with the ground truth report. However, it fails to mention the prolonged PR interval, which is a notable omission. Therefore, while the rhythm interpretation is partially accurate, it contains a significant error.
**Final Score: (0+0+5)/3 * 10 = 16.7**

---

Figure A11: Comparison of model outputs on ECG report generation task (Example 1). Blue indicates correct information, while red highlights errors. Our model's output fully aligns with the ground truth, with a report score of 10. In comparison, GPT-4's report, though structurally sound, contains notable inaccuracies despite its initial appearance of relevance.

---

## Case Study: PTB-XL Report

**Question:**
Please write a clinical report based on this ECG image.

**Ground Truth Report:**
Sinus rhythm. T waves are low or flat in V3, V4, V5 and slightly inverted in V2. Non-specific but consistent with myocardial ischemia. The age of the changes is uncertain.

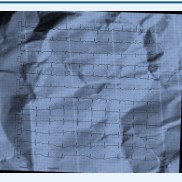

**PULSE Response:** The electrocardiogram (ECG) reveals a **normal sinus rhythm**, characterized by a regular heartbeat with a rate between 60-100 beats per minute, indicating that the heart's natural pacemaker, the sinoatrial (SA) node, is functioning properly. The axis of the heart's electrical activity is also within normal limits, suggesting that the electrical impulses are being conducted through the heart in a normal and coordinated manner. However, the ST segment and **T waves exhibit abnormalities, which may be indicative of myocardial ischemia or strain**, particularly in the anterior region of the heart. This could be a sign of reduced blood flow to the heart muscle, potentially leading to damage or even necrosis of the myocardial tissue. The presence of these abnormalities in the ST segment and T waves suggests that the heart is under stress, which may be caused by a variety of factors, including coronary artery disease, hypertension, or cardiomyopathy. Overall, the diagnosis based on this ECG report is **sinus rhythm** with ST changes, indicating that the heart is functioning normally in terms of rhythm, but is **exhibiting signs of potential myocardial ischemia or strain**.

**LLM-as-Judge Feedback for PULSE Response:**
**1. Diagnosis**
**Score: 8.** Explanation: The generated report correctly identifies the sinus rhythm and the presence of abnormalities in the ST segment and T waves, which are consistent with myocardial ischemia. However, it does not specifically mention the low or flat T waves in V3, V4, V5, and the slightly inverted T wave in V2, which are key details in the ground truth report.
**2. Form**
**Score: 7.** Explanation: The generated report accurately identifies the presence of ST segment and T wave abnormalities, which are indicative of myocardial ischemia. However, it lacks the specific details about the T wave morphology in V2, V3, V4, and V5, which are crucial for a precise comparison with the ground truth report.
**3. Rhythm**
**Score: 10.** Explanation: The generated report correctly identifies the sinus rhythm and provides a clear and accurate description of the heart's rhythm, matching the ground truth report.
**Final Score: (8+7+10)/3 * 10 = 83.3**

**GPT-4o Response:**
ECG Interpretation:
1. Heart Rate: The heart rate is approximately 75 beats per minute, which is within the normal range (60-100 bpm).
2. Rhythm: The rhythm appears to be regular with consistent R-R intervals, indicating a **normal sinus rhythm**.
3. P Waves: P waves are present before each QRS complex, indicating normal atrial depolarization. The P wave morphology is normal.
4. PR Interval: The PR interval is within the normal range (120-200 ms), suggesting normal atrioventricular conduction.
5. QRS Complex: The QRS duration is within the normal range (less than 120 ms), indicating normal ventricular depolarization. There are no significant Q waves observed, which rules out previous myocardial infarction.
6. ST Segment: The ST segments are isoelectric (flat) in all leads, indicating no signs of acute ischemia or injury.
7. T Waves: **T waves are upright in leads I, II, V3-V6, and inverted in aVR, which is a normal finding. There are no signs of T wave inversion or hyperacute T waves.**
8. QT Interval: The QT interval appears to be within the normal range (less than 440 ms for males and less than 460 ms for females), indicating no prolonged QT syndrome.
9. Axis: The QRS axis is normal, approximately 0 to +90 degrees.
10. Additional Observations: No evidence of left or right ventricular hypertrophy. No signs of bundle branch block or other conduction abnormalities.
Conclusion: The ECG findings are within normal limits. There are **no signs of acute ischemia, infarction**, or other significant cardiac abnormalities. The patient has a **normal sinus rhythm** with normal atrioventricular and intraventricular conduction.

**LLM-as-Judge Feedback for GPT-4o Response:**
**1. Diagnosis**
**Score: 0.** Explanation: The generated report fails to identify the key diagnosis of myocardial ischemia indicated by the T wave abnormalities in the ground truth report. It incorrectly concludes that there are no signs of acute ischemia or other significant cardiac abnormalities.
**2. Form**
**Score: 0.** Explanation: The generated report does not mention the low or flat T waves in V3, V4, V5, and the slightly inverted T waves in V2, which are critical waveform abnormalities noted in the ground truth report. Instead, it incorrectly states that T waves are upright in V3-V6 and does not acknowledge any T wave inversion.
**3. Rhythm**
**Score: 10.** Explanation: The generated report correctly identifies the rhythm as normal sinus rhythm, which is consistent with the ground truth report.
**Final Score: (0+0+10)/3 * 10 = 33.3**

Figure A12: Comparison of model outputs on ECG report generation task (Example 2). Blue indicates correct information, while red highlights errors. Our model's output mostly aligns with the ground truth, achieving a report score of 83.3. In comparison, GPT-4's output correctly identifies only the ECG rhythm, omitting most other key details.

---

**Case Study: PTB XL Report**

**Question:**
Please write a clinical report based on this ECG image.

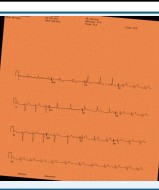

**Ground Truth Report:**
sinus arrhythmia, left axis deviation, left anterior hemiblock, right bundle branch block, bifascicular block, abnormal QRS(T), possible lateral myocardial damage

---

**PULSE Response:** The electrocardiogram (ECG) reveals a **sinus arrhythmia**, characterized by an irregular heart rhythm with varying intervals between beats, originating from the sinoatrial node. This arrhythmia is often benign and can be influenced by factors such as respiration, with heart rate increasing during inhalation and decreasing during exhalation. The ECG also demonstrates an **overdriven left type**, indicating that the left ventricle is dominant in the cardiac conduction system, which can be a normal variant or associated with certain cardiac conditions.
Furthermore, the ECG shows evidence of **left anterior hemiblock**, a conduction defect affecting the anterior fascicle of the left bundle branch, resulting in delayed activation of the anterior wall of the left ventricle. This can be a sign of underlying cardiac disease or injury. Additionally, the presence of **right bundle branch block** is noted, indicating a conduction defect affecting the right bundle branch, leading to delayed activation of the right ventricle. The combination of left anterior hemiblock and RBBB is consistent with **bifascicular block**, a condition where two of the three main fascicles of the cardiac conduction system are affected. The ECG also reveals a **left axis deviation**, indicating that the electrical axis of the heart is shifted to the left, which can be a normal variant or associated with certain cardiac conditions such as left ventricular hypertrophy or cardiomyopathy.
Overall, the ECG findings are consistent with a diagnosis of **left axis deviation, sinus arrhythmia, left anterior fascicular block, and right bundle branch block**.

---

**LLM as Judge Feedback for PULSE Response:**
**1. Diagnosis**
**Score: 7.** Explanation: The generated report correctly identifies sinus arrhythmia, left axis deviation, left anterior hemiblock, and right bundle branch block. It also correctly identifies the combination of left anterior hemiblock and RBBB as bifascicular block. However, it fails to mention 'abnormal QRS(T)' and 'possible lateral myocardial damage,' which are key diagnoses.
**2. Form**
**Score: 5.** Explanation: The generated report accurately describes the conduction defects (left anterior hemiblock, right bundle branch block) and left axis deviation. However, it does not address 'abnormal QRS(T)' and 'possible lateral myocardial damage,' which are critical waveform abnormalities mentioned in the ground truth report.
**3. Rhythm**
**Score: 10.** Explanation: The generated report provides a detailed and accurate description of sinus arrhythmia, including its characteristics and potential influences. This matches the rhythm interpretation in the ground truth report.
**Final Score: (7+6+10)/3 * 10 = 73**

---

**GPT 4o Response**: Clinical Report: Electrocardiogram (ECG) Analysis:
1. Heart Rate and Rhythm: Heart Rate: Approximately 75 beats per minute (bpm). Rhythm: **Sinus rhythm is present**, as indicated by the presence of P waves before each QRS complex and a consistent PR interval.
2. Intervals and Durations: PR Interval: 152 ms (within normal range: 120 200 ms). QRS Duration: 125 ms (slightly prolonged; normal range: 80 120 ms). QT Interval: 421 ms (QT interval, QTcB: 447.5 ms, QTcF: 439.0 ms; within normal limits for QTc).
3. Axis: QRS Axis: **72.0 degrees (left axis deviation).** P Axis: 71.5 degrees. T Axis: 73.0 degrees.
4. Waveform Analysis: P Waves: Normal morphology, consistent in all leads. QRS Complexes: Slightly widened, indicating a possible intraventricular conduction delay. ST Segment: No significant elevation or depression noted. T Waves: Normal morphology, no inversion or significant abnormalities observed.
5. Additional Observations: Lead V1 V6: Normal R wave progression. No evidence of acute ischemia or infarction. No significant arrhythmias detected.
Conclusion:
The ECG shows a **sinus rhythm with a heart rate of approximately 75 bpm**. There is a slight prolongation of the QRS duration, suggesting a possible intraventricular conduction delay. The QRS axis is deviated to the left. No acute ischemic changes or significant arrhythmias are noted. The overall findings are within normal limits, with minor deviations that may warrant further clinical correlation and follow up.

---

**LLM as Judge Feedback for GPT 4o Response:**
**1. Diagnosis**
**Score: 5**. Explanation: The generated report correctly identifies left axis deviation but misses several key diagnoses present in the ground truth report, such as sinus arrhythmia, left anterior hemiblock, right bundle branch block, bifascicular block, abnormal QRS(T), and possible lateral myocardial damage. The generated report mentions a possible intraventricular conduction delay, which is not explicitly stated in the ground truth report.
**2. Form**
**Score: 5**. Explanation: The generated report correctly identifies the left axis deviation and slightly prolonged QRS duration. However, it fails to identify the abnormal QRS(T) and possible lateral myocardial damage mentioned in the ground truth report. The generated report also does not mention any abnormalities in the P-wave, ST changes, or T-wave, which are critical for a comprehensive waveform analysis.
**3. Rhythm**
**Score: 5**. Explanation: The generated report identifies a sinus rhythm but misses the sinus arrhythmia mentioned in the ground truth report. The generated report does not identify any significant arrhythmias, which is a notable omission given the ground truth report's mention of sinus arrhythmia.
**Final Score: (5+5+5)/3 * 10 = 50**

---

Figure A13: Comparison of model outputs on ECG report generation task (Example 3). Blue indicates correct information, while red highlights errors. Our model's output mostly aligns with the ground truth report, achieving a report score of 73. In comparison, GPT-4's output partially aligns with the ground truth report.

**Case Study: ECG Arena**

**Question:**
What can be inferred from the pattern of the qR complexes and rS complexes in the leads of this ECG image?
**Ground Truth Answer:**
The pattern of qR complexes in leads I and aVL, along with rS complexes in leads II, III, and aVF, is characteristic of a Left Anterior Fascicular Block (LAFB). In LAFB, the ECG typically shows small Q waves and tall R waves in leads I and aVL (qR complexes), and small R waves followed by deep S waves in leads II, III, and aVF (rS complexes). This pattern is a result of the altered conduction through the left anterior fascicle, which causes the electrical activity to spread upwards and leftwards in the heart, leading to these distinctive ECG findings.
**Follow-up Question:**
What is the key feature that supports the diagnosis in this ECG image?
**Ground Truth Answer:**
The left axis deviation (LAD) observed in the ECG is a key feature that supports the diagnosis of Left Anterior Fascicular Block (LAFB). LAD is indicated by the negative deflections in leads II, III, and aVF, and positive deflections in leads I and aVL. In LAFB, the abnormal conduction pathway leads to the electrical axis of the heart shifting towards the left, which manifests as a left axis deviation on the ECG. This deviation, combined with the specific patterns of qR complexes in leads I and aVL and rS complexes in leads II, III, and aVF, strongly suggests LAFB.

**PULSE Response to the First Question:**
The presence of qR complexes and rS complexes in the leads suggests a pattern of abnormal ventricular depolarization. **The qR complexes, which are small q waves followed by a larger R wave, indicate a possible conduction delay or block in the ventricular conduction system. The rS complexes, which are small R waves followed by a larger S wave, may indicate a pattern of abnormal ventricular activation, possibly related to the left anterior fascicular block.** The combination of these complexes suggests that the ventricular depolarization is not following a normal pattern, which can be indicative of underlying cardiac pathology.
**PULSE Response to the Follow-up Question:**
The key feature that supports the diagnosis in this ECG image is the presence of a **left anterior fascicular block**…The left anterior fascicular block is a specific type of conduction block that affects the left anterior fascicle of the left bundle branch, resulting in abnormal ventricular activation. This diagnosis is further supported by the **presence of left ventricular hypertrophy**, which can contribute to the development of conduction abnormalities.

**LLM-as-Judge Feedback for PULSE Response:**
**1. Accuracy**
**Score: 5.** Explanation: The model's response correctly identifies the presence of qR and rS complexes and associates them with a conduction abnormality, specifically a left anterior fascicular block (LAFB). However, it introduces inaccuracies by suggesting a general 'abnormal ventricular depolarization' and mentioning left ventricular hypertrophy, which is not part of the ground truth answer.
**2. Completeness**
**Score: 5.** Explanation: The model's response partially covers the essential aspects of ECG interpretation related to LAFB, such as the identification of qR and rS complexes. However, it misses critical details like the specific leads involved (I, aVL, II, III, aVF) and the explanation of left axis deviation (LAD) as a key feature supporting the diagnosis.
**3. Instruction Adherence**
**Score: 7.** Explanation: The model follows the instructions by addressing the pattern of qR and rS complexes and suggesting a diagnosis. However, it deviates slightly by introducing unrelated information about left ventricular hypertrophy and not explicitly mentioning left axis deviation (LAD) as the key feature supporting the diagnosis.
**Final Score: (5+5+7)/3 * 10 = 56.7**

**GPT-4o Response to the First Question:**
The ECG image shows a pattern of qR complexes in leads V1 and V2, and rS complexes in leads V5 and V6. **This pattern is indicative of a right bundle branch block (RBBB).** Here are the key features that support this diagnosis:
1. qR Complexes in V1 and V2: In RBBB, the initial part of the QRS complex is normal because the left ventricle is activated first. This is followed by a delayed right ventricular activation, which results in a secondary R wave (R') in the right precordial leads (V1 and V2). This can appear as a qR complex. 2. rS Complexes in V5 and V6: In the left precordial leads (V5 and V6), the delayed right ventricular activation results in a wide S wave….In summary, the pattern of qR complexes in V1 and V2, and rS complexes in V5 and V6, along with a prolonged QRS duration, **is characteristic of a right bundle branch block.**
**GPT-4o Response to the Follow-up Question:**
The key feature that supports the **diagnosis of right bundle branch block (RBBB)** in this ECG image is the presence of a qR complex in leads V1 and V2 and an rS complex in leads V5 and V6. **Additionally, the QRS complex duration is prolonged, which is another hallmark of RBBB.**
To summarize: qR Complex in V1 and V2: This indicates delayed right ventricular activation…These features collectively support the **diagnosis of right bundle branch block.**

**LLM-as-Judge Feedback for GPT-4o Response:**
**1. Accuracy**
**Score: 0.** Explanation: The model's response is largely inaccurate. It incorrectly identifies the pattern of qR complexes in leads V1 and V2 and rS complexes in leads V5 and V6 as indicative of a right bundle branch block (RBBB). The ground truth specifies that the pattern of qR complexes in leads I and aVL, along with rS complexes in leads II, III, and aVF, is characteristic of a Left Anterior Fascicular Block (LAFB). The model's response does not match the correct identification of waveforms, intervals, and clinical diagnosis.
**2. Completeness**
**Score: 0.** Explanation: The model's response partially covers the essential aspects of ECG interpretation related to LAFB, such as the identification of qR and rS complexes. However, it misses critical details like the specific leads involved (I, aVL, II, III, aVF) and the explanation of left axis deviation (LAD) as a key feature supporting the diagnosis.
**3. Instruction Adherence**
**Score: 5.** Explanation: The model follows the instructions by addressing the pattern of qR and rS complexes and suggesting a diagnosis. However, it deviates slightly by introducing unrelated information about left ventricular hypertrophy and not explicitly mentioning left axis deviation (LAD) as the key feature supporting the diagnosis.
**Final Score: (0+0+5)/3 * 10 = 16.7**

Figure A14: Comparison of model outputs on ECG Arena (Example 1). Blue indicates correct information, while red highlights errors. Given the challenging nature of this task, our model's output partially aligns with the ground truth, and GPT-4o's output largely deviates from the reference.

# I  DISCUSSION

**Further improvement on model's instruction-following and multistep reasoning capabilities.**
To improve the model's capabilities of instruction-following and multistep reason, future work can focus on two main areas: (1) incorporating a more diverse set of instruction-following data to enhance the model's generalizability, and (2) scaling up high-quality chain-of-thought (CoT) and multi-turn training data informed by clinicians' expertise, established knowledge databases (e.g., SNOMEDCT (Stearns et al., 2001)), literature or textbooks. This curated data would include intermediate reasoning steps such as identifying key features, relating these features to diagnoses, and providing well-grounded rationales to enhance multistep reasoning.

We believe that scaling up and diversifying training data will improve instruction-following and multistep reasoning performance. This is also supported by our data ablation studies presented in Tables 5 and 6, which indicate the potential for improving model performance with additional training resources. We aim to explore these directions in future research to address the gaps noted in our current work.

