# OpenReview forum: "Teach Multimodal LLMs to Comprehend Electrocardiographic Images"
_ICLR.cc/2025/Conference — ICLR 2025 Conference Withdrawn Submission_

### Official Review · Reviewer_FYMk · 2024-10-27

**Soundness:** 3
**Presentation:** 3
**Contribution:** 2
**Rating:** 5
**Confidence:** 5

**Summary:**

This work proposes an impressive framework for instruction-tuning ECG images, evaluates it across multiple datasets, and introduces the first benchmark for ECG image instruction tuning.

**Strengths:**

- The first work to treat ECG signals as visual signals rather than time series.
- Builds the first benchmark for ECG instruction tuning with image input.

To the best of the reviewer's knowledge, this is the first work converting ECG signals into visual input, allowing ECG analysis to benefit from the advancements in the Visual Language Model (VLM) community. Additionally, the authors establish a comprehensive benchmark for evaluating ECG image instruction tuning, opening a new domain for ECG analysis.

**Weaknesses:**

Even though this work is well-executed and evaluated on multiple datasets, it still has some weaknesses:

- **Reliance on Accuracy and LLM Scoring**: The evaluation metrics are based solely on accuracy or LLM scoring. Incorporating human evaluation could enhance benchmark quality, as current LLMs are not specialized for ECG-related text understanding. I suggest using human evaluation for all open-ended tasks.

- **ViT Model Limitations**: The Vision Transformer (ViT) model is frozen, and while LLAVA uses ViT, it is not specifically tuned for ECG image features. This may degrade performance on ECG-specific tasks.

- **Translation Issues in Report Generation**: For report generation, the PTB-XL dataset originally includes German, Danish, and other European languages, not English. Translating these reports into English could lead to inaccuracies or loss of meaning.

- **Evaluation Metrics in Report Generation**: In the report generation task, only GPT is used as the LLM judge, without any lexical or clinical efficacy metrics. Clinical efficacy metrics are commonly used in other report generation tasks, such as in [1].

- **Relevance of Abnormality Detection as an LLM Task**: Is abnormality detection a necessary task for an LLM? According to Section 3.2, this task appears similar to classification. Additionally, LLMs can generate synonyms for disease names, raising the question of how to handle variations in terminology.


[1]Tanida, Tim, et al. "Interactive and explainable region-guided radiology report generation." Proceedings of the IEEE/CVF Conference on Computer Vision and Pattern Recognition. 2023.

**Questions:**

See weaknesses above.

---

> ### Author Response · Authors · 2024-11-23
> **Response to Reviewer FYMk (Part 1)**
>
> Thanks to the reviewer for the positive feedback on **the first work** converting ECG signals into visual input and establishing a **comprehensive benchmark** for evaluating ECG, opening **a new domain for ECG analysis.**
>
> > Reliance on Accuracy and LLM Scoring
>
> We conducted a human evaluation on 50 sampled reports from the PTB-XL Report and 30 questions from the ECG Arena. Human evaluation scores and their Pearson correlation with LLM-based scores are presented in Table 1. The results indicate a strong correlation between the two sets of scores. In the final version, we will extend this evaluation to the full dataset and include comprehensive human evaluation results.
> | Models | PTB-XL Report (LLM score) | PTB-XL Report (Human score) | Pearsons  Correlation | ECG Arena  (LLM score) | ECG Arena  (Human score) | Pearsons  Correlation |
> |--------|---------------------------|-----------------------------|-----------------------|------------------------|--------------------------|-----------------------|
> | GPT-4o | 51.9                      | 50.8                        | 93.4                  | 32.8                   | 35.1                     | 92.0                  |
> | PULSE  | 62.8                      | 64.1                        | 91.9                  | 37.4                   | 39.0                     | 91.7                  |
>
> <Table 1> Comparison of human evaluation scores and LLM-based scores on report generation and ECG arena
>
> > ViT Model Limitations
>
> Thanks for pointing this out! We conducted an ablation study by unfreezing the ViT module parameters during training and reported the model performance in Table 2 below. The results show a performance improvement (i.e., average score from 71.8 to 75.0) compared to the original model with frozen ViT parameters. **We have included these new findings in the revision.**
> | Models       | PTB-XL Super | PTB-XL Report | CSN  | CODE-15 | ECQ-QA | CPSC | G12  | MMMU ECG | ECG Arena | AVG  |
> |--------------|--------------|---------------|------|---------|--------|------|------|----------|-----------|------|
> | Unfrozen ViT | **76.9**         | **65.4**          | **87.9** | **87.0**    | 71.6   | **65.4** | **81.4** | **64.0**     | **41.5**      | **75.0** |
> | Frozen ViT        | 74.8         | 61.3          | 85.2 | 85.4    | **73.8**   | 57.6 | 78.2 | 58.0     | 38.9      | 71.8 |
>
> <Table 2> Ablation results on the impact of unfreezing ViT module parameters
>
> > Translation Issues in Report Generation
>
> We acknowledge the importance of maintaining accurate translations, particularly in evaluation tasks. During the evaluation data curation process, we validated the translated reports for 500 test samples through two methods: 1) manual verification by human experts and 2) automated validation, where the translated reports are compared against SCP-ECG (standardized protocols for ECG analysis including rhythms, waveforms, and diagnoses) provided along with each ECG in the dataset [Wagner et al.].
>
> Wagner, Patrick, et al. "PTB-XL, a large publicly available electrocardiography dataset." Scientific data 7.1 (2020): 1-15.
>
> > Evaluation Metrics in Report Generation
>
> We include clinical efficacy metrics to evaluate the model performance in report generation. Specifically, we label the generated reports and compare them with ground truths across 23 categories (i.e., diagnostic subclass in PTB-XL). Metrics including precision, recall, and F1-score are utilized for evaluation. Since no established labeling tool exists for ECG reports, we leverage a language model (GPT-4o) to extract labels based on the 23 categories. We also experimented with a rule-based labeling approach, but it performed poorly in this context.  As shown in Table 3 below, our model significantly outperforms representative proprietary and open-source MLLMs in terms of clinical efficacy metrics, highlighting its robustness across diverse evaluation criteria.
> |                   | P        | R        | F1       |
> |-------------------|----------|----------|----------|
> | GPT-4o            | 7.1      | 5.7      | 4.8      |
> | Claude 3.5 Sonnet | 10.2     | 9.1      | 8.5      |
> | Qwen2-VL-72B      | 4.3      | 4.6      | 4.0      |
> | LLaVA-Med         | 1.5      | 4.4      | 0.3      |
> | PULSE             | **28.4** | **32.1** | **28.9** |
>
> <Table 3> Comparison results on report generation with respect to clinical efficacy metrics.

---

> > ### Author Response · Authors · 2024-11-23
> > **Response to Reviewer FYMk (Part 2)**
> >
> > > Relevance of Abnormality Detection as an LLM Task
> >
> > Abnormality detection is a critical task for assessing a model’s ability to understand and interpret ECG images similarly to human experts. This task has been extensively studied in ECG interpretation, as timely and accurate identification of abnormalities is essential for effective heart disease management in clinical practice [Ribeiro et al.].  While abnormality detection can be framed as a classification task or other question formats for easier benchmarking of LLMs, the most accurate and clinically meaningful definition of this task remains "abnormality detection".
> >
> > Ribeiro, Antônio H., et al. "Automatic diagnosis of the 12-lead ECG using a deep neural network." Nature communications 11.1 (2020): 1760.
> >
> > > LLMs can generate synonyms for disease names, raising the question of how to handle variations in terminology.
> >
> > For closed-ended and multiple-choice questions, the model generates answers based on the provided options, minimizing the impact of terminology variation issues. For open-ended questions, we implemented terminology mappings (following [Khunte et al.]), which link diverse terminologies to their standardized equivalents. This approach ensures correct answers are recognized, even when the model employs alternative wording.
> >
> > Khunte, Akshay, et al. "Automated Diagnostic Reports from Images of Electrocardiograms at the Point-of-Care." medRxiv (2024).

---

> > > ### Comment · Reviewer_FYMk · 2024-11-24
> > >
> > > I agree that anomaly detection is important for ECG analysis. However, it is not an appropriate application for an LLM-based method, as LLMs are primarily designed for text generation.
> > >
> > > I will maintain my score due to the improper claim and insufficient supporting results.

---

> > > > ### Author Response · Authors · 2024-11-24
> > > >
> > > > We appreciate the reviewer’s observation and would like to clarify potential misunderstandings:
> > > >
> > > > 1. **Clarification on “Abnormal Detection” vs. “Anomaly Detection”**: It seems there might be some confusion in terminology. Our study focuses on **abnormal detection**, which refers to identifying **clinically significant abnormalities in ECG** readings, such as arrhythmias or ischemic changes, based on established medical guidelines. This is fundamentally different from **anomaly detection**, which typically refers to identifying statistical outliers or deviations from a norm.
> > > > 2. **Rationale for Using LLMs**: The adoption of **LLMs as the backbone for our study is grounded in their unique ability to generalize across diverse tasks through instruction tuning**. Unlike domain-specific models, which require task-specific engineering, LLMs leverage their training across a broad spectrum of tasks to adapt to new scenarios with minimal additional effort. This capability is particularly advantageous in ECG analysis, which encompasses varied tasks such as abnormality detection, report generation, and multi-turn interaction in diagnostic reasoning.
> > > >
> > > > 3. **Unified Framework for ECG Tasks**: Our approach formulates these varied tasks—whether they involve abnormal detection, generating textual summaries of ECG findings, or engaging in multi-turn diagnostic challenges—as prompt-based instructions. This **unification allows the LLM to operate flexibly across a wide range of applications within ECG analysis, eliminating the need for separate specialized models for each task**. Such versatility is one of the key strengths of LLMs, making them a suitable and novel tool for addressing challenges in this domain.
> > > >
> > > > By integrating LLMs, we aim not just to address specific tasks but to create a generalizable, instruction-driven framework for comprehensive ECG analysis. We hope this explanation resolves the concerns about the applicability of LLMs in this context and highlights the rationale for their use in our study.

---

> > ### Comment · Reviewer_FYMk · 2024-11-24
> >
> > Thank you for the detailed rebuttal.
> >
> > From Table 3, if I understand correctly, the results shown are zero-shot from other MLLMs, whereas PULSE uses 1.2M data for SFT, resulting in better performance. While this improvement is straightforward, can it truly be considered the main contribution? The claim that a model fine-tuned (SFT) on specific domain data achieves higher performance than a general-domain model is expected and not particularly novel.
> >
> > I believe the community would benefit from a comparison of results where models like Qwen-2VL or LLaVA-Med are fine-tuned on the same dataset used for PULSE.

---

> > > ### Author Response · Authors · 2024-11-24
> > >
> > > We appreciate the reviewer’s feedback and would like to clarify the main contributions of our work, as the focus extends beyond demonstrating the effectiveness of supervised fine-tuning (SFT), which is indeed a well-established practice.
> > >
> > > 1. **Novelty in ECG Multimodal Learning**: As the reviewer themselves also highlighted, there are currently no general or medical MLLMs capable of effectively interpreting ECG images. Our study is **the first to explore how to construct a large-scale, effective training dataset for ECG-specific instruction-tuning in this context**.
> > >
> > > 2. **Development of a Domain-Specific Dataset**: **The core novelty of our work lies in creating a large-scale, clinician-inspired, comprehensive, and diverse ECG instruction-tuning dataset**. This dataset is developed through a scalable, cost-efficient approach, addressing a critical gap as no similar dataset currently exists in this domain. This point has also been positively noted by Reviewers 198c and zTNk.
> > >
> > > 3. **Broader Applicability**: We emphasize that the dataset is not limited to improving the performance of a single genera MLLM. It is designed to be **a valuable resource for fine-tuning any general-purpose MLLMs** (e.g., Qwen-2VL-7B, LLaVA, etc.) to enhance their capabilities in ECG image interpretation. **The user could fine-tune any general MLLMs in Table 3 and we believe our dataset would be useful to boost the models’ performance on ECG tasks.**
> > >
> > > 4. **Evaluation of Generalization**: The inclusion of general MLLMs in Table 3 serves to highlight two points:
> > >    - Current general-purpose MLLMs perform poorly in ECG-related tasks, which highlights the need for this study.
> > >    - Fine-tuning these MLLMs with our dataset significantly boosts their performance, demonstrating the utility of the dataset and methodology.
> > >
> > >    Furthermore, our evaluation is not limited to in-domain datasets. By testing on highly out-of-domain (OOD) datasets, we demonstrate the generalizability of the model, reinforcing that our approach does not merely overfit to specific datasets but contributes to broader ECG interpretation tasks.
> > >
> > > 5. **Future Comparisons**: Again, our primary goal is to establish the foundation by creating the dataset and showing its effectiveness though we already added Qwen2-VL fine-tuning results in the response to Reviewer 198c.
> > >
> > > | Models                       | PTB-XL Super | PTB-XL Report | CSN  | G12  | MMMU ECG |
> > > |------------------------------|--------------|---------------|------|------|----------|
> > > | Qwen2-VL-7B                  | 22.4         | 43.0          | 25.5 | 32.9 | 31.5     |
> > > | LLaVA-v1.6-Vicuna-7B         | 15.8         | 16.5          | 23.7 | 23.3 | 28.0     |
> > > | PULSE (Qwen2-VL-7B)          | 73.1         | 58.6          | 82.5 | 75.9 | 54.5     |
> > > | PULSE (LLaVA-v1.6-Vicuna-7B) | 74.8         | 61.3          | 85.2 | 78.2 | 58.0     |
> > >
> > > We hope this clarifies that our main contributions lie in addressing a critical unmet need in the domain, rather than the SFT process itself, and we appreciate the reviewer’s engagement.

---

### Official Review · Reviewer_AoRP · 2024-10-29

**Soundness:** 2
**Presentation:** 3
**Contribution:** 1
**Rating:** 3
**Confidence:** 4

**Summary:**

The paper proposes a fine-tuned multi-modal LLM tailored for ECG image interpretation. Additionally, the paper introduces a large ECG image instruction tuning dataset with 1 million examples, covering a wide range of ECG-related tasks from diverse data sources. A new evaluation benchmark for ECG image interpretation tasks is also proposed.

**Strengths:**

The paper's strengths include:

- many SOTA MLLMs as baselines
- Evaluation on a wide range of tasks
- Preparation of a benchmark that could be used in future works. This may be relevant for the field.
- Preparation of an instruction-tuning dataset that could be used in future works. Again, this may be relevant for the field.

**Weaknesses:**

The weaknesses include:

- Motivation: while there may be cases where only printouts of ECGs are available, the need for AI models on ECG printouts is still questionable.
- Related works: Current multimodally trained ECG models are neither discussed nor compared against, e.g.:
Radhakrishnan et al. (2023), "Cross-modal autoencoder framework learns holistic representations of cardiovascular state", https://www.nature.com/articles/s41467-023-38125-0, Turgut et al. (2023), "Unlocking the Diagnostic Potential of ECG through Knowledge Transfer from Cardiac MRI", https://arxiv.org/pdf/2308.05764
- Current foundational time series models are neither discussed nor compared against, e.g.: Yang et al. (2023), "BIOT: Biosignal Transformer for Cross-data Learning in the Wild", https://proceedings.neurips.cc/paper_files/paper/2023/hash/f6b30f3e2dd9cb53bbf2024402d02295-Abstract-Conference.html, - It would have been interesting to see how much using the ECG-images instead of the raw ECGs limits the performance.
- An ablation study using an ECG encoder instead of the image model would be interesting
- An ablation study using the image encoder directly on ECG images, without the LLMs would be interesting
- To provide robustness guarantees, have the authors evaluated their methods as well as the baselines across multiple seeds set during fine-tuning?Goswami et al. (2024), "MOMENT: A Family of Open Time-series Foundation Models", https://arxiv.org/pdf/2402.03885
- Statements such as "limitations of traditional ECG models" (ll 64-65) are made without elaborating and providing references.
- Experiments / Evaluation: For fair comparison, Table 3 and 4 should include the ECG baselines mentioned above. This would highlight the differences between imaging and time series modality with respect to downstream performance.
- Furthermore, domain-specific methods in Table 3 and 4 have not been tuned for the reported downstream tasks. The authors make the effort to tune multiple MLLMs on the applications under investigation, however, they only report scores from the original paper for the domain-specific methods.
- In Table 4, MERL achieves best performance on the CPSC 2018 task, however, the authors mistakenly highlight their own method PULSE as the best performing one.
- In Table 3 and 4, the authors report that certain methods are not applicable or not designed for certain tasks, which is why they opt to not report scores in those cases. However, for the open-source MLLMs that are not designed for ECG image analysis either, the authors tune the models and report performance scores. This clearly is an unfair comparison, further highlighting the need for tuning the domain-specific methods and reporting their performance.
- All LLM-based baselines have not been trained on ECGs / ECG-images, while ECG baselines are not able to handle all tasks. So for some of the tasks (the text-based ones) no meaningful baseline is provided. While such baselines may not be available, the quality of the model could still be compared with strong baselines by splitting it into subtasks, i.e. use established standard models to extract relevant features from a given ECG (heart rate, irregularities, abnormalities, …). Then include the extracted features in a prompt and ask a general or medical LLM to provide an answer. This could handle most of the tasks and thus provide a meaningful baseline.
- Limited novelty: This seems to be a very applied paper without any novelty in the method (note that the provided instruction tuning and evaluation datasets are still relevant for the field)

**Questions:**

- Are there many use cases, where ECGs are not available in electronic form, but where scanners and AI models are available and could be integrated?
- Even if only printouts are available, is directly interpreting them as images the best solution? Wouldn’t the more reliable solution be to convert these images into electronic ECGs first (the problem seems relatively simply due to the high contrast of the plots and often clearly visible reference lines) and the use established ECG models?
- Motivation lacks reasoning: Why would the authors transform readily available ECG recordings, e.g. 800k samples from MIMIC-ECG, into images before analysing them? The majority of the transformed image represents background, i.e. noise. Is the patch projector of the imaging encoder (ViT) capable of learning a meaningful signal from the data?
- It would have been interesting to see how much using the ECG-images instead of the raw ECGs limits the performance.
- An ablation study using an ECG encoder instead of the image model would be interesting
- An ablation study using the image encoder directly on ECG images, without the LLMs would be interesting
- To provide robustness guarantees, have the authors evaluated their methods as well as the baselines across multiple seeds set during fine-tuning?

---

> ### Author Response · Authors · 2024-11-23
> **Response to Reviewer AoRP (Part 1)**
>
> Thanks to the reviewer for acknowledging our contributions, including the **instruction-tuning dataset** covering a **wide range** of ECG-related tasks and a **new evaluation benchmark for ECG image interpretation**, both of which advance the field!
>
> > The need for AI models on ECG printouts
>
> The need for AI models capable of analyzing ECG printouts is highlighted by the potential to overcome limitations of signal-based models, expand access to advanced diagnostic tools in resource-constrained settings, seamlessly integrate with existing clinical workflows, and support diverse applications such as retrospective studies, remote healthcare, and AI-assisted cardiology. See our detailed illustrations below.
> * **Limitation of current signal-based models.** In many clinical settings, ECG data are frequently stored only as printed images without corresponding signal repositories, particularly in resource-constrained, and remote settings [Siontis et al.,]. These environments often face limited access to specialized medical expertise, while the increasingly accumulated paper ECGs surpass the capacity of human experts to analyze them effectively [Schopfer et al.]. However, these widely used paper ECGs or scanned ECG images are incompatible with existing ECG models, which are trained and tested exclusively on time-series physiological signals. Furthermore, variations in data formats and system architectures among ECG device vendors [Cuevas-González et al.] pose additional challenges to the interoperability and practical application of signal-based models in clinical practice [Chung et al.,].
> * **Motivation for developing image-based models.** Image-based models can address the aforementioned limitations by enabling AI inference directly from ECG images, which are the primary formats used by clinicians [Cuevas-González et al.,]. This approach accommodates rural, low-resource, or remote clinic settings, where signals may be unavailable, making advanced diagnostic tools accessible to broader populations.
> * **Clinical implementation for image-based models.** In many healthcare facilities, access to inexpensive portable scanners (or even smartphone cameras) is far more common than access to advanced ECG signal management systems. AI tools for ECG images can be implemented on lightweight hardware, such as laptops or mobile devices, with minimal computational requirements. Clinicians already rely heavily on printed ECGs for diagnosis, making image-based AI solutions a natural extension of current practices without requiring systemic changes.
> * **Multiple real-world use cases.** Image-based models offer diverse real-world applications, including 1) supporting remote clinics and low-resource healthcare where ECGs are printed for interpretation and long-term storage lacks infrastructure for signal storage or processing [Chung et al.,]; 2) enabling retrospective studies by analyzing archived ECG printouts; 3) AI-assistant for cardiology. With the ability to comprehend and respond to complex natural language queries, the models can be used as AI assistants for cardiology, enhancing human-in-the-loop clinical decision-making, education, and research.
>
> Siontis, Konstantinos C., et al. "Artificial intelligence-enhanced electrocardiography in cardiovascular disease management." Nature Reviews Cardiology 18.7 (2021): 465-478.
>
> Schopfer, David W. "Rural health disparities in chronic heart disease." Preventive Medicine 152 (2021): 106782.
>
> Cuevas-González, Daniel, et al. "ECG standards and formats for interoperability between mHealth and healthcare information systems: a scoping review." International Journal of Environmental Research and Public Health 19.19 (2022): 11941.
>
> Chung, Cheuk To, et al. "Clinical significance, challenges and limitations in using artificial intelligence for electrocardiography-based diagnosis." International journal of arrhythmia 23.1 (2022): 24.
>
> > Additional comparison with existing signal-based ECG models
>
> We evaluate our model against MMCL and MOMENT under three different settings: zero-shot, linear probing, and full fine-tuning. In the zero-shot setting, MMCL and MOMENT extract ECG signal features and compute the cosine similarity with the text features of class names, which serves as the classification probability. The text encoders used are all-MiniLM-L6-v2 and bert-base-uncased, chosen to match the feature dimensions of the respective models. We also attach a linear classification head to the three models, evaluating them in other two settings: freezing the encoder and fine-tuning only the classification head (Linear Prob) and jointly training both the encoder and the classification head (Full Finetune).

---

> > ### Author Response · Authors · 2024-11-23
> > **Response to Reviewer AoRP (Part 2)**
> >
> > > Additional comparison with existing signal-based ECG models (continue)
> >
> > As shown in Table 1 below, the baseline models only achieve higher performance than our model on the in-domain evaluation dataset (PTB-XL). However, they exhibit significantly lower performance on out-of-domain datasets, demonstrating a lack of generalization compared to our model. Notably, signal-based models face limitations in real-world scenarios, as they cannot be applied to datasets consisting solely of digital images and/or open-ended questions (e.g., ECG Arena).
> >
> > | Model  | PTB-XL: In-domain (Zero-shot/Linear Prob/Full Finetune) | CSN: Out-of-domain | CPSC: Out-of-domain | ECG Arena: Out-of-domain |
> > |--------|---------------------------------------------------------|--------------------|---------------------|--------------------------|
> > | MMCL   | 54.3/81.6/85.3                                          | 12.5               | 52.7                | N/A                      |
> > | Moment | 49.8/**88.3**/83.3                                      | 24.8               | 50.5                | N/A                      |
> > | PULSE  | 82.4                                                    | **85.2**           | **76.9**            | 38.9                     |
> >
> > <Table 1> Comparisons of our model with signal-based ECG models. AUC scores are reported for PTB-XL and CPSC, and accuracy is reported for CSN. N/A indicates that signal-based models are not applicable to tasks involving only digital images and open-ended questions.
> >
> > > Ablation study using an ECG encoder instead of the image model
> >
> > Thanks for pointing this out. Due to the significant training time required, this experiment is still running. We will update the results as soon as they are available.
> >
> > > Ablation study using the image encoder directly on ECG images, without the LLMs
> >
> > We compare our model against a variant that adopts the same image encoder as ours but without incorporating LLMs. Similar to the signal-based ECG model experimental setup, we evaluate the performance under three settings (zero-shot, linear prob, and full finetune) on both in-domain and out-of-domain datasets. As shown in Table 2 below, this image encoder-only model achieves comparable performance to those signal-based baselines (as reported in Table 1 above) and even better performance on PTB-XL. However, it still faces significant generalizability limitations. While the image encoder is capable of processing ECG images, the model still cannot handle diverse tasks (e.g., answering open-ended questions in ECG Arena).
> >
> > | Model                     | PTB-XL: In-domain (Zero-shot/Linear Prob/Full Finetune) | CSN: Out-of-domain | CPSC: Out-of-domain | ECG Arena: Out-of-domain |
> > |---------------------------|---------------------------------------------------------|--------------------|---------------------|--------------------------|
> > | Image-encoder only | 49.9/66.1/**90.3**                                      | 18.3               | 46.2                | N/A                      |
> > | PULSE                     | 82.4                                                    | **85.2**           | **76.9**            | 38.9                     |
> >
> > <Table 2> Comparisons of our model with the image-encoder-only model without using LLMs
> >
> > > Evaluation of model robustness across multiple random seeds
> >
> > Given the training cost constraints and significant improvements our model demonstrated over the baselines, we evaluated our model across 20 random seeds during testing and provided the averaged results along with standard deviations in Table 3 below. These results highlight the robustness of our model across different seeds.
> > | Models     | PTB-XL Super | PTB-XL Report | CSN       | CODE-15   | ECQ-QA    | CPSC      | G12       | MMMU ECG  | ECG Arena |
> > |------------|--------------|---------------|-----------|-----------|-----------|-----------|-----------|-----------|-----------|
> > | PULSE(std) | 75.5(1.3)    | 62.1(1.2)     | 84.5(0.9) | 84.1(1.5) | 72.1(1.1) | 57.9(1.2) | 77.6(0.7) | 58.5(0.4) | 39.3(0.6) |
> >
> > <Table 3> Evaluation results across 20 random seeds. The average and standard deviation (std) are reported.

---

> > > ### Author Response · Authors · 2024-11-23
> > > **Response to Reviewer AoRP (Part 3)**
> > >
> > > > Limitations of traditional ECG models
> > >
> > > The application of traditional ECG models in real-world clinical settings encounters several challenges. First, these models are primarily designed for classification tasks with limited cardiac conditions (e.g., 6 types of abnormalities [Ribeiro et al.,]), often lacking generalizability. Second, they typically treat ECG data as time-series physiological signals, which may not always be available, particularly in resource-constrained or remote settings [Siontis et al., 2021]. In such settings, ECG data are often stored exclusively as printed or digital images [Sangha et al., 2022; 2023], limiting the utility of signal-based models. Third, variations in data formats across different device vendors [Cuevas-Gonz´alez et al., 2022] further complicate the interoperability and applicability of traditional ECG models in diverse healthcare environments [Chung et al., 2022]. **We have revised the manuscript accordingly with additional supporting references**.
> > >
> > > Ribeiro, Antônio H., et al. "Automatic diagnosis of the 12-lead ECG using a deep neural network." Nature communications 11.1 (2020): 1760.
> > >
> > > > Fair comparison among open-source MLLMs and ECG models
> > >
> > > Our comparisons are fair across open-source MLLMs and domain-specific models. For open-source MLLMs, we utilize zero-shot prompting **without task-specific fine-tuning**. These models leverage their comprehensive pretraining on diverse datasets, including visual instruction data, to naturally adapt to new tasks [Liu et al.]. In contrast, domain-specific models are typically designed for single, specialized tasks (e.g., multi-label classification within predefined categories). While these specialized models may achieve superior performance in their targeted domains (e.g., diagnostic classification), they typically struggle with broader tasks like report generation or handling complex natural language queries. Thus, the comparison remains fair as neither open-source MLLMs nor domain-specific models are fine-tuned for the evaluations.
> > >
> > > Liu, Haotian, et al. "Visual instruction tuning." Advances in neural information processing systems 36 (2024).
> > >
> > > > Comparison with strong baselines by splitting it into subtasks (ECG feature extraction + prompting)
> > >
> > > We compared our model with the reviewer’s proposed strong baselines (Table 4 below). For PTB-XL, we extracted relevant features such as heart rate, axes, waveform intervals, amplitudes, onsets, and offsets, based on PTB-XL+ [Strodthoff et al.], which includes annotations from both commercial and open-source ECG analysis tools. These features were then used to prompt GPT-4o to generate responses. However, this approach exhibited relatively low performance compared to our model.
> > >
> > > We also assessed another baseline that employed an established ECG model, SE-WRN [Zhong et al.], as a feature extractor. This method extracted features and potential disease diagnoses, which were further refined using prediction probability scores to ensure high-quality inputs before being converted into prompt text. Despite these enhancements, this approach achieved a performance of only 51.6 on the ECG-QA dataset, falling short of our model’s performance.
> > >
> > > These findings demonstrate that even with advanced feature extraction and carefully constructed descriptions, directly prompting current LLMs is insufficient for accurate ECG interpretation. This is primarily because current LLMs lack the ability to understand either the original ECG images/time-series signals or the processed features necessary for precise interpretations. These limitations highlight the **significant gap in current LLM capabilities in analyzing ECGs** and support the motivation and novelty of our work: **teaching MLLMs to comprehend ECGs**.
> > >
> > > | Models         | PTB-XL Super | ECQ-QA |
> > > |----------------|--------------|--------|
> > > | Feature Prompt | 35.6         | 51.6*  |
> > > | PULSE          | **74.8**         | **73.8**   |
> > >
> > > <Table 4> Comparisons with feature extraction and text prompting methods. * score is copied from ECG-QA paper [Oh et al.,]
> > >
> > > Strodthoff, Nils, et al. "PTB-XL+, a comprehensive electrocardiographic feature dataset." Scientific data 10.1 (2023): 279.
> > >
> > > Zhong, Xian, et al. "Squeeze-and-excitation wide residual networks in image classification." 2019 IEEE International Conference on Image Processing (ICIP). IEEE, 2019.
> > >
> > > Oh, Jungwoo, et al. "Ecg-qa: A comprehensive question answering dataset combined with electrocardiogram." Advances in Neural Information Processing Systems 36 (2024).

---

> > > > ### Author Response · Authors · 2024-11-23
> > > > **Response to Reviewer AoRP (Part 4)**
> > > >
> > > > > Novelty
> > > >
> > > > We present the first comprehensive study on using MLLMs for interpreting ECG images, addressing critical limitations in existing methodologies and introducing a standardized benchmark for model evaluation in this domain.
> > > > * **The use of MLLMs for ECG Image Interpretation.** The problem we studied is novel that, to the best of our knowledge, we are the first work to investigate the MLLMs in ECG image interpretation (also mentioned by reviewer *FYMk*). Compared to existing ECG models that are trained and tested on time-series ECG for specific tasks, our model can directly read printed or scanned ECG images and provide interpretations given complex user queries for a variety of ECG-related tasks.
> > > > * **Addressing the Gap in Current MLLMs for ECG Interpretation.** Despite the importance of interpreting ECG images (e.g., application in resource-constrained or remote settings, complexities in clinical implementations, etc.), even state-of-the-art MLLMs (e.g., GPT-4o, Gemini-1.5 Pro, or Claude 3.5 Sonnet) fail to provide accurate and comprehensive ECG interpretations, as demonstrated in Figure 1. Our work identifies this critical gap between the capabilities of current methodologies and real-world clinical needs. We aim to bridge this gap by teaching MLLMs to comprehend ECG images in a manner akin to human interpretation.
> > > > * **Methodology for instruction-tuning data curation.** Training such models is challenging due to the lack of large-scale, high-quality, and diverse datasets comprising ECG images and associated text instructions. Our data curation process stands out through 1) **realistic image synthesis** resembling the real-world artifacts (e.g., distortions, varying background colors, injected textual meta-information, etc.) in paper ECGs to enhance the model robustness against “imperfect” images commonly encountered in practice; 2) **incorporating clinician’s insights** to identify key ECG-related tasks for more effective understanding; 3) **diverse instruction generation**: generate comprehensive and diverse instruction data based on original diagnosis and/or clinician’s reports using language models in a scalable approach; 4) **quality checking**: filtering out low-quality data to maintain high standards for training.
> > > > * **Comprehensive evaluation for benchmarking and future directions.** Evaluation is as crucial as model development, especially in the context of LLMs. A robust evaluation suite like ECGBench not only benchmarks model performance but also highlights current limitations, guiding future advancements. As discussed in the paper, ECGBench provides a concrete assessment framework for identifying gaps (e.g., enhancing multistep and complex reasoning capabilities) and informing the next steps in improving model capabilities.
> > > >
> > > > > Even if only printouts are available, is directly interpreting them as images the best solution?
> > > >
> > > > While converting ECG printouts into signal data might appear to be a promising alternative, directly interpreting them as images presents a more practical and robust solution for several key reasons:
> > > > * **Practicality and technical challenges in signal reconstruction.** Despite the high contrast and reference lines in ECG images, reconstruction techniques often struggle with artifacts such as scanner-induced distortions, variable lighting conditions, and smudges on printouts. These artifacts can introduce inaccuracies into the reconstructed signals, significantly reducing their diagnostic utility. Even if ECG images are converted back into electronic signals, the resulting signal data may still differ significantly from the high-fidelity signals that signal-based models are trained on. On the other hand, natively processing the ECGs as images avoids these intermediate reconstruction inaccuracies. Notably, in our paper, we enhance robustness by synthesizing realistic ECG images that emulate the common artifacts found in printed or scanned ECGs, enabling our model to perform reliably in real-world settings.
> > > > * **Limited generalizability of established signal-based models.** Even assuming perfect reconstruction, existing signal-based ECG models often exhibit limited generalizability. These models are typically tailored to specific tasks (e.g., classification of certain diseases). In contrast, our MLLM paradigm, trained on ECGInstruct, is designed to generalize across diverse tasks beyond classification, enabling it to tackle a wide range of use cases (e.g., generating clinical reports, and interpreting key features, AI assistant with human-in-the-loop decision-making, etc.).
> > > > * **Broader applicability and flexibility of image-based models.** Image-based models have inherent versatility and can be applied to diverse real-world scenarios beyond signal analysis. For example, they can natively interpret text annotations, grid markers, and other contextual information on ECG printouts, which signal-based models cannot leverage.

---

> > > > > ### Author Response · Authors · 2024-11-23
> > > > > **Response to Reviewer AoRP (Part 5)**
> > > > >
> > > > > > Why would the authors transform readily available ECG recordings, e.g. 800k samples from MIMIC-ECG, into images before analysing them?
> > > > >
> > > > > Our primary goal is to develop MLLMs capable of interpreting ECG images, as these are the most common cost-efficient storage formats, especially in rural and remote settings. Transforming ECG recordings into synthetic paper-like ECG images provides a **scalable and practical** solution to generate **large-scale** training datasets containing ECG image-text pairs.
> > > > >
> > > > > The background in ECG images is not merely noise; it also provides consistent contextual cues, such as gridlines and axes, that help structure the visual input. These elements are important in analyzing waveforms, as human experts often rely on gridlines to assess morphological abnormalities. Through training, the model aligns image patches with corresponding text instructions, enabling it to learn to focus on important areas within the images. In addition, our experiments demonstrate that the model achieves outstanding performance across multiple evaluation datasets, validating the effectiveness of the ViT module in encoding ECG images.
> > > > >
> > > > > > Minors: In Table 4, MERL achieves best performance on the CPSC 2018 task
> > > > >
> > > > > We have fixed this in the revision.

---

> > > > > ### Comment · Reviewer_AoRP · 2024-11-24
> > > > >
> > > > > Thank you for your reply. I am not convinced by the responses. In particular, while the problem that is studied here has not been studied before, I am still not convinced that this problem formulation makes sense. In particular, the issues quoted for the practicality and technical challenges in signal reconstruction seem rather artificial, many of them will also apply to image-based interpretation. I also find it rather questionable that any contextual information on ECG printouts would be beneficial.

---

> > > > > > ### Author Response · Authors · 2024-11-27
> > > > > > **Comparison between image-based encoder and signal-based encoder**
> > > > > >
> > > > > > We present the comparison results between the image-based encoder and the signal-based encoder in the table below. Both models were trained using the same data and architectural framework, differing only in the ECG encoding approach. The results show that the **image-based encoder consistently outperforms the signal-based encoder** across all datasets, with particularly significant improvements observed in out-of-domain datasets (e.g., CPSC). These findings highlight that encoding ECGs as images not only **aligns with the goal of enabling broader applicability of automated ECG diagnosis especially in resource-constrained or remote settings** (where only printed or digital ECGs are available) but also **empirically surpasses the performance of signal-based encoder model**.
> > > > > >
> > > > > > | Models               | PTB-XL Super | PTB-XL Report | CSN  | CODE-15 | ECQ-QA | CPSC | G12  | MMMU ECG | ECG Arena |
> > > > > > |----------------------|--------------|---------------|------|---------|--------|------|------|----------|-----------|
> > > > > > | Signal encoder       | 67.1         | 55.5          | 80.6 | 82.6    | 63.3   | 32.5 | 71.6 | NA       | NA        |
> > > > > > | Image encoder (PULSE) | **74.8**         | **61.3**          | **85.2** | **85.4**    | **73.8**   | **57.6** | **78.2** | **58.0**     | **38.9**      |

---

### Official Review · Reviewer_zTNk · 2024-11-02

**Soundness:** 3
**Presentation:** 3
**Contribution:** 3
**Rating:** 8
**Confidence:** 3

**Summary:**

This paper introduces PULSE, a multimodal large language model (MLLM) tailored for interpreting electrocardiogram (ECG) images, addressing the limitations of existing methods which often rely on raw physiological signals and lack generalizability. The authors contribute a new dataset, ECGInstruct, comprising over one million ECG image-text samples to fine-tune PULSE, and ECGBench. Experiments demonstrate that PULSE outperforms both proprietary and open-source MLLMs, with an average accuracy improvement of 15% to 30%.

**Strengths:**

This work contributes a comprehensive ECG image instruction tuning dataset from diverse data sources, which facilitates the advancement of MLLMs in understanding ECG images.
The proposed PULSE model surpassed proprietary and open-source MLLMs by a large range in diverse tasks related to ECG image comprehension. The model, data and code have been released.

**Weaknesses:**

As stated in the discussion, the dataset contains few multistep instructions, which could undermine the multistep reasoning capability of the PULSE model and limits the report generation performance.

**Questions:**

According to table 4, the performance gain in arena score which reflects the model’s instruction-following ability is limited compared with Claude 3.5 Sonnet in arena score. The authors could include more discussion about the results to help improve multistep reasoning capabilities in future research.

---

> ### Author Response · Authors · 2024-11-23
> **Response to Reviewer zTNk**
>
> Thanks to the reviewer for positive feedback on constructing **comprehensive ECG image instruction tuning** dataset from diverse data sources and **outstanding model performance** against proprietary and open-source MLLMs **by a large range** in diverse tasks.
>
> > Further improvement in instruction-following and multistep reasoning capabilities
>
> To improve the model’s capabilities of instruction-following and multistep reason, future work can focus on two main areas: (1) incorporating a more diverse set of instruction-following data to enhance the model's generalizability, and (2) scaling up high-quality chain-of-thought (CoT) and multi-turn training data informed by clinicians' expertise, established knowledge databases (e.g., SNOMEDCT [Stearns et al.,]), literature or textbooks. This curated data would include intermediate reasoning steps such as identifying key features, relating these features to diagnoses, and providing well-grounded rationales to enhance multistep reasoning.
>
> We believe that scaling up and diversifying training data will improve instruction-following and multistep reasoning performance. This is also supported by our data ablation studies presented in Tables 5 and 6, which indicate the potential for improving model performance with additional training resources. We aim to explore these directions in future research to address the gaps noted in our current work. **We have included this discussion in the revision.**
>
> Stearns, Michael Q., et al. "SNOMED clinical terms: overview of the development process and project status." Proceedings of the AMIA Symposium. American Medical Informatics Association, 2001.

---

### Official Review · Reviewer_198c · 2024-11-04

**Soundness:** 3
**Presentation:** 3
**Contribution:** 3
**Rating:** 6
**Confidence:** 4

**Summary:**

This study presents the creation of a fine-tuning dataset and benchmark for ECG in multi-language language models (MLLM), which appears to hold value for advancing medical AI. The dataset is synthetically generated with various distortion noises, and Llama 3 was used for ECGInstruct and quality checking; however, the accuracy of these implementations has not been assessed, raising concerns about potential LLM bias. Therefore, evaluation on independent downstream task datasets would be essential.

**Strengths:**

The study’s strengths include the development of a large-scale dataset and the introduction of extensive benchmark tasks, both of which contribute positively to the field.

**Weaknesses:**

The study lacks a strong technical novelty and technical details.
For a fairer comparison, it would be preferable to include fine-tuning results against other LLMs.
Additionally, human accuracy should be evaluated, at least partially, for the ECGBench tasks.
It is also recommended to assess performance on external validation tasks beyond the ECGBench dataset.

**Questions:**

Please provide technical details.
For a fairer comparison, it would be preferable to include fine-tuning results against other LLMs.
Additionally, human accuracy should be evaluated, at least partially, for the ECGBench tasks.
It is also recommended to assess performance on external validation tasks beyond the ECGBench dataset.

---

> ### Author Response · Authors · 2024-11-23
> **Response to Reviewer 198c (Part 1)**
>
> Thanks to the reviewer for recognizing our **contributions**, including the curated training dataset **ECGInstruct** and the comprehensive benchmarking suite **ECGBench**, to the field!
>
> > Technical novelty
>
> We present the first comprehensive study on using MLLMs for interpreting ECG images, addressing critical limitations in existing methodologies and introducing a standardized benchmark for model evaluation in this domain.
> * **The use of MLLMs for ECG Image Interpretation.** The problem we studied is novel that, to the best of our knowledge, we are the first work to investigate the MLLMs in ECG image interpretation (also mentioned by reviewer *FYMk*). Compared to existing ECG models that are trained and tested on time-series ECG for specific tasks, our model can directly read printed or scanned ECG images and provide interpretations given complex user queries for a variety of ECG-related tasks.
> * **Addressing the Gap in Current MLLMs for ECG Interpretation.** Despite the importance of interpreting ECG images (e.g., application in resource-constrained or remote settings, complexities in clinical implementations, etc.), even state-of-the-art MLLMs (e.g., GPT-4o, Gemini-1.5 Pro, or Claude 3.5 Sonnet) fail to provide accurate and comprehensive ECG interpretations, as demonstrated in Figure 1. Our work identifies this critical gap between the capabilities of current methodologies and real-world clinical needs. We aim to bridge this gap by teaching MLLMs to comprehend ECG images in a manner akin to human interpretation.
> * **Methodology for instruction-tuning data curation.** Training such models is challenging due to the lack of large-scale, high-quality, and diverse datasets comprising ECG images and associated text instructions. Our data curation process stands out through 1) **realistic image synthesis** resembling the real-world artifacts (e.g., distortions, varying background colors, injected textual meta-information, etc.) in paper ECGs to enhance the model robustness against “imperfect” images commonly encountered in practice; 2) **incorporating clinician’s insights** to identify key ECG-related tasks for more effective understanding; 3) **diverse instruction generation**: generate comprehensive and diverse instruction data based on original diagnosis and/or clinician’s reports using language models in a scalable approach; 4) **quality checking**: filtering out low-quality data to maintain high standards for training.
> * **Comprehensive evaluation for benchmarking and future directions.** Evaluation is as crucial as model development, especially in the context of LLMs. A robust evaluation suite like ECGBench not only benchmarks model performance but also highlights current limitations, guiding future advancements. As discussed in the paper, ECGBench provides a concrete assessment framework for identifying gaps (e.g., enhancing multistep and complex reasoning capabilities) and informing the next steps in improving model capabilities.
>
> > Technical details
>
> We follow the model architecture of LLaVA, which includes three core components: a vision encoder, a large language model, and a projector to align image and text modalities. Table 1 below summarizes all the model parameters. Specifically, for the LLM, we utilize Vicuna-1.5-7B, while the vision encoder is based on CLIP-ViT-Large-Patch14-336. We employ a 2-layer MLP as a projector to map the visual features from the CLIP encoder onto the tokens used by the LLM. These features are mapped onto predefined image tokens, which encapsulate the features of ECG images. The tokens representing ECG features are then concatenated as an image context preceding the dialogue.
>
> We format all datasets into a chatbot-style multi-turn dialogue format (same as Vicuna-1.5-7B) and use the special token `<image>` to represent image features within the text data. For example, a sample data instance is: “Human: <image> Describe this ECG image.\nAssistant: This image …”. To enhance the model’s ability to handle ECG images of various sizes encountered in real-world scenarios, we employ Anyres. Anyres divides high-resolution images into multiple sub-images of size 336x336. The features of these sub-images are then concatenated with the global features of the original image to form the final image representation.
>
> We fine-tune all parameters of the vision encoder (see newly added vision encoder unfrozen experiments in response to the reviewer FYMk <Table 2>), projector, and the LLM. The training process uses a learning rate of 2e-5, a batch size of 128, and a cosine scheduler with a 5% warm-up period over three epochs. The loss is calculated using the cross-entropy loss function, focusing on the response portion of the dialogue. **We have included the technical details and Table 1 in the revision.**

---

> > ### Author Response · Authors · 2024-11-23
> > **Response to Reviewer 198c (Part 2)**
> >
> > > Technical details (continue)
> >
> > | Model Parameters                           |        |
> > |--------------------------------------------|--------|
> > | Total                                      | 7.06B  |
> > | Vision Encoder(clip-vit-large-patch14-336) | 303.5M |
> > | Connector                                  | 21M    |
> > | LLM(Vicuna-1.5-7B)                         | 6.74B  |
> > | Training Parameters                        |        |
> > | Learning Rate                              | 2e-5   |
> > | Weight Decay                               | 0.0    |
> > | Warmup Ratio                               | 0.03   |
> > | Learning Rate Scheduler                    | Cosine |
> > | Batch Size                                 | 128    |
> > | Vision Encoder Arch                        |        |
> > | Hidden Size                                | 1024   |
> > | Input Resolution                           | 336    |
> > | ViT Layer                                  | 24     |
> > | ViT Heads                                  | 16     |
> > | Patch Size                                 | 14     |
> > | LLM Arch                                   |        |
> > | Hidden Size                                | 4096   |
> > | Max Context Length                         | 4096   |
> > | Attention Heads                            | 32     |
> > | Hidden Layers                              | 32     |
> > | KV Heads                                   | 32     |
> > <Table 1> Summary of model parameters.
> >
> > > Fine-tuning results against other LLMs
> >
> > We fine-tune Qwen2-VL-7B using ECGInstruct and show the results in Table 2 below. The performance of the two backbone models is comparable, with LLaVA slightly outperforming Qwen.
> > | Models                        | PTB-XL Super | PTB-XL Report | CSN  | G12  | MMMU ECG |
> > |-------------------------------|--------------|---------------|------|------|----------|
> > | PULSE (Qwen2-VL-7B)           | 73.1         | 58.6          | 82.5 | 75.9 | 54.5     |
> > | PULSE (LLaVA-v1.6 -Vicuna-7B) | 74.8         | 61.3          | 85.2 | 78.2 | 58.0     |
> > <Table 2> Comparison results of different MLLM backbones
> >
> > > Human accuracy
> >
> > We engaged three domain experts specializing in ECG to evaluate a sample of 30 questions from the MMMU ECG. The performance comparison is provided in Table 2 below. Our findings highlight a significant performance gap between current MLLMs and human expertise, indicating the need for further improvements in model capabilities for ECG image analysis. In the final version, we will include a more comprehensive evaluation of human performance and report the results to provide additional insights.
> > | Category               | MMMU ECG (%) |
> > |------------------------|--------------|
> > | Human Expert (Low)     | 70.0         |
> > | Human Expert (Medium)  | 90.0         |
> > | Human Expert (High)    | 93.3         |
> > | Human Expert (Average) | 84.4         |
> > | GPT-4o                 | 43.5         |
> > | PULSE                  | 58.0         |
> > <Table 2> Comparison results on MMMU ECG among human experts, GPT-4o, and PULSE
> >
> > > External validation
> >
> > Our ECGBench incorporates **external** (out-of-domain) evaluation datasets, as indicated in Table 2 by the last column labeled “In-Domain?” These datasets consist of data that was completely unseen during training. Notably, ECGBench features an **“in-the-wild”** evaluation setting (**MMMU ECG** and **ECG Arena**), where all images and questions were sourced entirely from **external** resources (e.g., textbooks, web, and literature) rather than being synthetically generated from existing ECG datasets.

---

### Note · Authors · 2025-01-28

I have read and agree with the venue's withdrawal policy on behalf of myself and my co-authors.